# Any-Property-Conditional Molecule Generation with Self-Criticism using Spanning Trees

**Alexia Jolicoeur-Martineau**                                   *alexia.j@samsung.com*
*Samsung SAIL Montréal*

**Aristide Baratin**                                             *a.baratin@samsung.com*
*Samsung SAIL Montréal*

**Kisoo Kwon**                                                   *kisoo.kwon@samsung.com*
*Artificial Intelligence Center, Device Solutions, Samsung Electronics*

**Boris Knyazev**                                                *b.knyazev@samsung.com*
*Samsung SAIL Montréal*

**Yan Zhang**                                                    *y2.zhang@samsung.com*
*Samsung SAIL Montréal*

**Reviewed on OpenReview:** *https://openreview.net/forum?id=QGZd5Bfb1L*

## Abstract

Generating novel molecules is challenging, with most representations of molecules leading to generative models producing many invalid molecules. Spanning Tree-based Graph Generation (STGG) (Ahn et al., 2021) is a promising approach to ensure the generation of valid molecules, outperforming state-of-the-art generative models for unconditional generation. In practice, it is desirable to generate molecules conditional on one or multiple target properties rather than unconditionally. Thus, we extend STGG to multi-property conditional generation. Our approach, **STGG+**, incorporates a modern Transformer architecture, random masking of properties during training (enabling conditioning on *any* subset of properties and classifier-free guidance), an auxiliary property-prediction loss (allowing the model to *self-criticize* molecules and select the best ones), and other improvements. We show that **STGG+** achieves state-of-the-art performance on in-distribution and out-of-distribution conditional generation, as well as reward maximization.

## 1 Introduction

Generating novel molecules is challenging, and the choice of molecular representation significantly impacts the performance of generative models. Recent methods generate molecules in 2D (as graphs) (Jo et al., 2022; Vignac et al., 2022; Thompson et al., 2022; Jo et al., 2023) or 3D (Hoogeboom et al., 2022; Bao et al., 2022; Xu et al., 2023; Huang et al., 2024). However, 1D representations, such as SMILES (Weininger, 1988), remain the standard representation in practical molecule design (Segler et al., 2018; Kwon et al., 2023; M. Bran et al., 2024; Wu et al., 2024).

A key issue with molecule representations is that a single error by the generative model can result in invalid molecules, which is especially challenging as the molecule size increases. Spanning Tree-based Graph Generation (STGG) (Ahn et al., 2021) tackle this issues by masking invalid tokens during sampling, preventing the generation of invalid structures (e.g., atoms without bonds between them, branch end before branch start) and ensuring proper valency. The masking employed by STGG not only prevents invalid molecules, but also leads to higher-quality and more diverse generated molecules, outperforming or matching more recent state-of-the-art generative models for unconditional generation (Jang et al., 2023). However, its application in multi-property conditional settings has not been explored. We address this setting along with a few additional challenges, as discussed below.

**Any-property-conditioning.** In real-world applications, it is desirable to generate molecules *conditional on one or multiple target properties* rather than unconditionally Jain et al. (2023). Furthermore, we want to condition on *different* subsets of desirable properties without needing to retrain the model for each subset.

**Self-criticism.** A critical issue in molecule discovery is synthesis time, which can take weeks or months. To avoid extreme synthesis costs, we need to filter the molecules that we provide to chemists. Some properties can be verified through simulations, but this is slow, and not all properties can be simulated. Another option is to rely on external property predictor models, but training, validating, and managing multiple property predictors can be challenging. To address this issue, we propose giving the model the ability to predict properties and thus *self-criticize* its own generated molecules, allowing it to filter out those with undesirable properties. Such self-criticism mechanisms are common in language models Madaan et al. (2024); however, they are not used in STGG.

**Out-of-distribution properties.** We often seek to generate novel molecules with out-of-distribution (OOD) properties to discover new structures. Classifier-Free Guidance (CFG) (Ho & Salimans, 2022) is a technique to improve conditioning fidelity; we found CFG useful for in-distribution properties, but problematic for some OOD conditioning values (especially extreme values), resulting in poor generated molecules. Since guidance can be beneficial to some conditioning values but not others, we propose *random guidance* with best-of-$k$ self-filtering (described in Section 3.5).

To summarize, we tackle any-property-conditional molecule generation in a practical, real-world setup. In doing so, we make the following contributions:

1. We make multiple improvements compared to STGG (Figure 1; ablation in Section 4.5):

    (a) **Any-property-conditioning** enabling conditioning on multiple target properties and any subset of them without retraining (Section 3.2);

    (b) **Improved architecture** based on recent advances in Transformers (Section 3.1);

    (c) **Improved Spanning-Tree** with enhanced tokenization and masking to improve validity and generalization (Section 3.3);

    (d) **Auxiliary property prediction objective** enabling the self-criticism ability and improving conditioning fidelity (Section 3.6);

    (e) **Random guidance for extreme value conditioning** to improve classifier-free guidance on extreme OOD conditioning fidelity (Section 3.5).

2. By comprehensive evaluation, we demonstrate excellent performance in terms of:

    (a) high quality and diversity for unconditional generation on QM9 and Zinc (Appendix A.7);

    (b) high quality and conditioning fidelity for in-distribution (Section 4.1) and OOD (Sections 4.2 and 4.4) conditional generation on the HIV, BACE, and BBBP datasets, as well as the challenging Chromophore DB with larger and more complex molecules;

    (c) high reward and diversity for multi-objective reward maximization on QM9 (Section 4.3).

Our evaluation spans diverse molecule generation scenarios, including realistic (yet underexplored in machine learning) cases like the Chromophore DB dataset and HOMO-LUMO property optimization. This underscores the significance of our proposed approach and its exploration.

## 2 Background and Related Work

### 2.1 1D vs 2D vs 3D representations

There are many ways of representing molecules in the context of molecular generation. Some of the most popular methods are (1) autoregressive models on 1D strings (Segler et al., 2018; Ahn et al., 2021; Kwon et al., 2023) and (2) diffusion (Song & Ermon, 2019; Ho et al., 2020; Song et al., 2020) models on 2D graphs. While both approaches have similar sample complexity, 1D strings offer a more compressed representation, requiring less space and fewer parameters and, thus, increased potential for scalability. We provide a detailed comparison between different representations in Appendix A.1.

Furthermore, recent results indicate that 1D strings are as competitive as 2D molecular graph methods for both unconditional molecule generation (Jang et al., 2023; Fang et al., 2023) and property prediction (Yüksel et al., 2023).

Both 1D and 2D representations encapsulate the same amount of information, making the choice largely a matter of preference. We advocate for 1D string representations due to their scalability and effective utilization by Transformer models, and thus, we focus on this type of representation in our work.

Molecules are inherently 3D objects and can be generated in 3D (Hoogeboom et al., 2022; Bao et al., 2022; Xu et al., 2023; Huang et al., 2024; Song et al., 2024; Luo et al., 2024; Verma et al., 2022; Mercatali et al., 2024). Generating 3D molecules is challenging and often results in fewer valid molecules. Since 3D conformations can be inferred from 1D or 2D molecular representations using popular chem-informatics tool Sun et al. (2020); Landrum et al. (2024), 1D strings like SMILES remain the standard representation in molecule design (M. Bran et al., 2024; Wu et al., 2024).

### 2.2   1D representations

### 2.2.1   SMILES

The most popular choice of 1D string-based representation is SMILES (Weininger, 1988), an extremely versatile method capable of representing any molecule. However, when used in generative models, generated SMILES strings often correspond to invalid molecules. A single incorrectly placed token often leads to an invalid molecule. Graph-based diffusion methods also face a similar issue (Vignac et al., 2022). Recent methods such as STGG(Ahn et al., 2021) and SELFIES (Krenn et al., 2022) were developed to tackle this issue by preventing the generation of invalid molecules. For a detailed comparison of SMILES, SELFIES, and STGG, see Appendix A.2.

### 2.2.2   SELFIES

SELFIES has been shown to perform worse than SMILES on property-conditional molecule generation (Gao et al., 2022; Ghugare et al., 2023). Meanwhile, SMILES has been shown to perform worse than STGG on unconditional generation using the same architecture (Ahn et al., 2021). Furthermore, STGG performs on-par or better than state-of-the-art unconditional generative models(Ahn et al., 2021; Jang et al., 2023). Therefore, our work focuses exclusively on extending and improving STGG.

### 2.2.3   STGG

Spanning Tree-based Graph Generation (STGG) (Ahn et al., 2021) is a type of generative model made to tackle the problem of invalid generated molecules. It tackle this issue by using a new type of representation, which allows it to easily mask invalid tokens during sampling, preventing the generation of invalid structures (e.g., atoms without bonds between them, branch end before branch start) and ensuring proper valency. STGG (Ahn et al., 2021) uses a vocabulary similar to SMILES but with some key differences: begin " (" and end ") " branch tokens, ring start "[bor]" and $i$-th ring end "[eor-$i$]" tokens. Contrary to SELFIES, STGG was made from the ground up for molecule generation. STGG leverages a Transformer architecture (Vaswani et al., 2017) to sample the next tokens conditional on the tokens of the current unfinished molecule. To predict the ring end tokens, STGG uses a similarity-based output layer distinct from the linear output layer used to predict other tokens. STGG also uses an input embedding to track the number of open rings. Invalid next tokens are prevented through masking of next tokens that would lead to impossible valencies (e.g., atoms, ring-start, and branch-start when insufficient valency remains) and structurally invalid tokens (e.g., ring-$i$ end when fewer than $i$ ring start tokens are present).

In the next section, we show how to improve STGG's architecture, vocabulary, and masking, adapt STGG for any-property conditional generation, and improve conditioning fidelity through classifier-free guidance, self-criticism, and random classifier-free guidance for extreme conditioning.

## 3   Method

### 3.1   Architecture

We enhance the architecture used in STGG, which is a regular Transformer architecture (Vaswani et al., 2017), directly from PyTorch main libraries. The original transformer consists of alternating blocks of Multilayer perceptron (MLP) and self-attention layers. To improve it, we leverage recent improvements in Large Language Models following GPT-3 (Radford et al., 2019), Mistral (Jiang et al., 2023), and Llama (Touvron et al., 2023). The improvements include:

1. bias-free architecture (Chowdhery et al., 2023), which consists of removing the bias/intercept from the linear layers in order to reduce the number of parameters and stabilize training

2. replacing replace Layer Normalization (LayerNorm) (Ba et al., 2016) with Root Mean Square Layer Normalization (RMSNorm) (Zhang & Sennrich, 2019), which consists in replacing z-score normalization of the layers by a simpler un-centered version (which is equivalent to removing the bias term in the LayerNorm).

3. residual-path weight initialization (Radford et al., 2019), which ensures that the variance of the input is equal to the variance of the output (instead of having the variance collapse to 0 at the output, leading to vanishing gradients)

4. rotary embeddings (Su et al., 2024) instead of relative positional embedding to encode token position because it improves generalization

5. Flash-Attention-2 (Dao et al., 2022; Dao, 2023) which is a fused GPU kernel for a faster and more memory-efficient GPU implementation of self-attention.

6. SwiGLU (Hendrycks & Gimpel, 2016; Shazeer, 2020), which is an improvement to the original MLP with special gating and Swish activation function.

7. changes in hyper-parameters following GPT-3 (Radford et al., 2019) (i.e., AdamW (Loshchilov & Hutter, 2017; Kingma & Ba, 2014) $\beta_2 = 0.95$, cosine annealing schedule (Loshchilov & Hutter, 2016), more attention heads, no dropout).

These modifications aim to enhance the model's efficiency, scalability, and overall performance. We also considered more efficient architectures (Appendix A.5).

## 3.2 Any-property conditioning

To condition the model on any subset of desired properties without retraining, we need to be able to "turn off" properties at inference time. To do this, we include additional masking features that indicate when a conditioning variable is missing or not: whenever a variable is missing, we reset it to a default value and set the corresponding binary "missing" feature to 1. For variables that are present, the binary "missing" feature is 0. The binary feature is needed for the model to be able to distinguish between a variable being turned off for conditioning and the variable being present with that default value. Otherwise, the model would frequently be conditioned incorrectly: when two properties X and Y are correlated but only Y is masked, then the model would learn that they are decorrelated due to Y being set to the default value *independently* of X. With the mask, the model can learn that the value of the Y variable is irrelevant instead.

For continuous variables, we choose the default value to be 0. We make sure to $z$-score standardize all continuous features first, so that the default corresponds to the mean. For categorical variable, we can include the binary masking feature in the variable itself by adding an extra category that indicates that it is missing (a variable with two categories [A, B] becomes [A, B, missing]). With one-hot encoding of the categories, this amounts to setting all the original categories to 0 and the binary mask to 1.

During training, we mask a random subset of $t$ properties, where $t$ is chosen uniformly between 0 and the total number of properties (Appendix A.3.2). This lets the model see many combinations at train time and thus lets us condition the model on any subset of desired properties at test time.

Our model takes the standardized continuous features (continuous properties concatenated with their binary missing indicators) as input and processes them using a 2-layer multilayer perceptron (MLP) with the Swish activation (Hendrycks & Gimpel, 2016; Ramachandran et al., 2017). Each categorical feature is then processed individually using an embedding layer. These processed outputs are added directly to the embedding of all tokens (Figure 1).

## 3.3 Improvements to Spanning-Tree

Starting from STGG as base, we implement several improvements. Firstly, we extend the vocabulary to allow for the generation of molecular compounds that are composed of multiple unconnected graphs (e.g., salt is represented as [Na+].[Cl-], where [Na+] and [Cl-] are single-atom molecules connected through an ionic bond). STGG uses a fixed

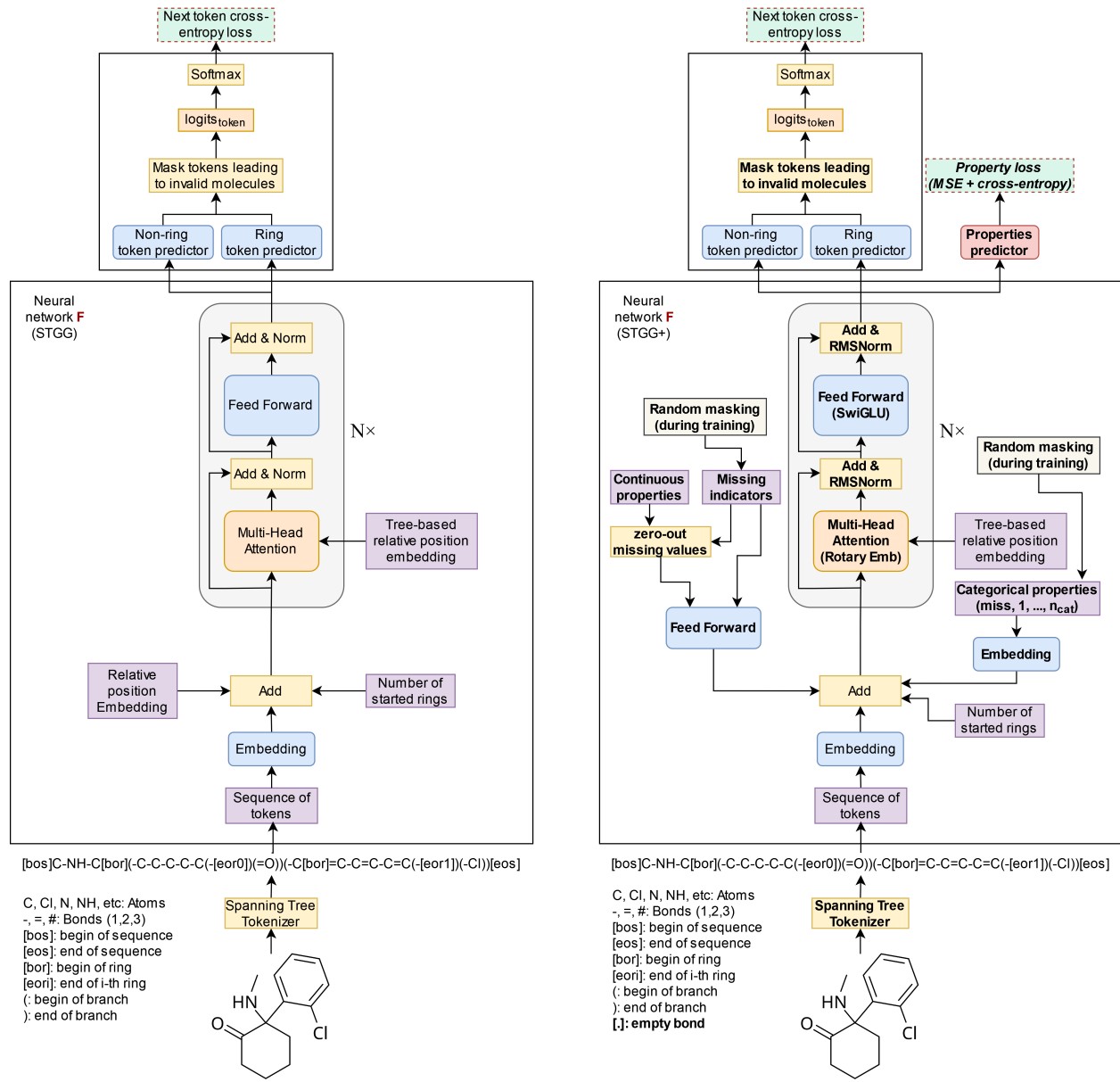

Figure 1: Left: STGG architecture, Right: Our **STGG+** architecture. The molecule is tokenized and embedded. Properties embeddings are added. The output produces the property and next-token predictions (masked to prevent invalid tokens). Novel components compared to STGG are in **bold**. See Section 3 for explanations on the components. See Section 4.5 for an ablation showing the importance of each component.

vocabulary and a fixed set of maximum valencies that determines how many valence bonds each atom can form. Instead of requiring a predefined vocabulary, we automate the process of building a vocabulary based on the atoms found in the dataset and their maximum valency, again derived from the dataset. This data-centric approach allows us to represent complex structures, such as hypervalent molecules (molecules with more than 8 valence electrons).

We observe that STGG can occasionally generate incomplete samples by creating too many branches without closing them within the allowed maximum length, particularly when conditioning on extreme out-of-distribution properties (above or below 4 standard deviations). To address this, we modify the token masking process to ensure the model closes its branches when the number of open branches approaches the number of tokens left to reach the maximum

length. This additional masking step prevents the rare but problematic situation of incompletely-generated samples. Additionally, for larger molecules, it is possible for the model to produce more rings than the maximum number of rings (100); we now mask the ring-start token when the maximum number is reached. With these additional masks, we maintain near perfect validity, even when generating molecules with out-of-distribution properties. We describe the masking algorithms in details (before and after the changes) in Appendix A.12.

Contrary to STGG, we do not canonicalize molecules and instead use a random ordering of molecule components (a different random ordering is sampled for each molecule during training). This improves generalization as it can be seen as data augmentation (see ablations in Section 4.5).

### 3.4    Classifier-free guidance

To enforce better conditioning of the properties, we use classifier-free guidance (CFG), originally designed for diffusion models (Ho & Salimans, 2022), which was found to be beneficial for autoregressive language models too (Sanchez et al., 2023). This technique involves directing the model more toward the conditional model's direction while pushing it away from the unconditional model's direction by an equal amount (this is best visualized in Figure 2; the softmax sampling equation shows that when $w > 1$, we push toward the conditional model and away from the unconditional model).

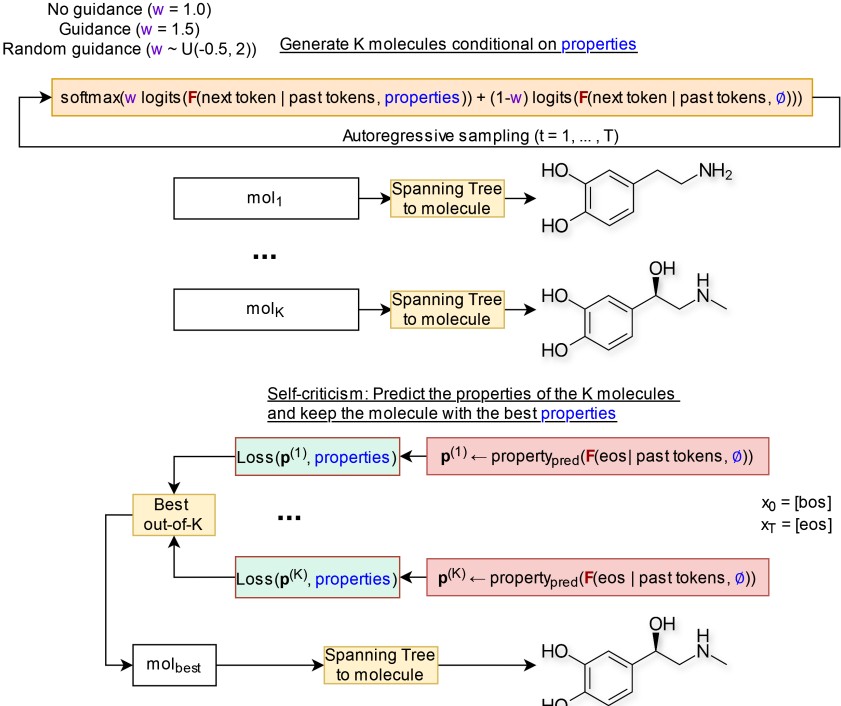

Figure 2: Generation and self-prediction using **STGG+**. Generate $K$ molecules conditional on properties using classifier-free guidance. The unconditional model predicts the properties of the $K$ molecules; the molecule assumed closest to the desired properties is returned.

### 3.5    Random guidance for extreme conditioning

Guidance (Ho & Salimans, 2022) can be problematic for extreme (out-of-distribution) conditioning values, resulting in poor "generative efficiency" (% of valid, unique, and novel molecules) and conditioning fidelity. However, guidance can still be beneficial to some extreme conditioning values. To improve generative performance on extreme conditioning values, we propose to randomly sample a guidance $w \sim \mathcal{U}(-0.5, 2)$ for each sample, ensuring high diversity through a mix of low and high guidance. Then, using self-criticism, our method selects the best-out-of-$k$ molecule from the

molecules generated at different guidance levels, indirectly allowing the model to determine by itself which guidance is best for each sample. This is effectively a way to balance exploration and exploitation (higher guidance means less diversity and better property-alignment, while lower guidance means more diversity and less property-alignment).

### 3.6 Self-criticism

To make the model more powerful, we provide the model with the ability to self-criticize its own generated molecules. The purpose is improve the quality of generated samples by using a jointly-trained property predictor to rank and filter the generated samples. It works as follows: 1) the model generates $k$ molecules for a given set of properties, 2) it evaluates the $k$ molecules properties based on its own property-predictor (see the paragraph below), and 3) it returns the molecule whose properties best match the conditioned properties. This best-out-of-$k$ strategy improves the quality of its generated molecules.

For the model to be able to predict properties of the molecules, we add a property-prediction loss to the training objective (Figure 1). During training, the model is tasked with predicting both the next token and the properties of the current unfinished molecule. During sampling, we generate molecules conditioned on desired properties with classifier-free guidance (Figure 2). Then, we mask out the properties (making them as fully missing) and reprocess the molecule until we reach the end-of-sequence (EOS) token. At this point, we extract the predicted property of this molecule.

## 4 Experiments

We run four sets of experiments. First, we show that our model can generate molecules conditioned on properties from the test set with high fidelity. Second, we show that our model can efficiently generate (high % of novel, unique, and valid) molecules with high fidelity on out-of-distribution (OOD) properties. Third, we show that our model can produce molecules that maximize a reward function, achieving similar or better performance compared to online learning methods using offline learning. Finally, we show that our model can generate high fidelity molecules conditioned on out-of-distribution (OOD) properties on a small dataset of larger and more complex molecules.

We experiment with six datasets: (1) QM9 (Ramakrishnan et al., 2014) with around 134k molecules and maximum SMILES length of 37; (2) Zinc250K (Sterling & Irwin, 2015) with 250k molecules and maximum length of 136; (3) BBBP (Wu et al., 2018) with 862 molecules and maximum length of 186; (4) BACE (Wu et al., 2018) with 1332 molecules and maximum length of 161; (5) HIV (Wu et al., 2018) with 2372 molecules and maximum length of 193; (6) Chromophore DB (Joung et al., 2020) with 6810 molecules and maximum length of 511. See Appendix A.3 for details on these datasets, A.4 for more information on the hyperparameters, A.6 for property prediction performance metrics of the self-critic. We also perform unconditional generation experiments on QM9 and Zinc250k in Appendix A.7. We rely on the following software: PyTorch (Paszke et al., 2019), Molecular Sets (MOSES) (Polykovskiy et al., 2020) and RDKit (Landrum et al., 2024). Unless otherwise specified, we use RDKit to evaluate the properties of the generated molecules.

We follow the same protocol as Liu et al. (2024). We train our model on HIV, BACE, and BBBP. We use the same train, valid, and test splits as Liu et al. (2024). Each dataset has an experimental categorical property related to HIV virus replication inhibition (HIV), blood-brain barrier permeability (BBBP), or human $\beta$-secretase 1 inhibition (BACE), respectively, and two continuous properties: synthetic accessibility (SAS) (Ertl & Schuffenhauer, 2009) and complexity scores (SCS) (Coley et al., 2018). We evaluate the models using metrics on distribution and fidelity of conditioning after generating molecules conditional on properties from the test set. The condition control metrics are the Mean Absolute Error (MAE) of SAS (evaluated by RDKit) and accuracy of the categorical property (evaluated by a Random Forest (Breiman, 2001) predictor using the Morgan Fingerprint (Morgan, 1965; Gao et al., 2022)). The distribution metrics are validity, atom coverage in the largest connected graph (how many unique atom types are produced), internal diversity (average pairwise similarity of generated molecules), fragment-based similarity (Degen et al., 2008), Fréchet ChemNet Distance (FCD) (Preuer et al., 2018). We consider any atom coverage above the test set coverage to indicate good coverage. The Property accuracy metric depends on a RandomForest classifier, thus we consider any accuracy equal or above the test set to indicate good condition control.

## 4.1 In-distribution conditional generation

**Baselines.** Following Liu et al. (2024), we compare our method to strong recent baselines: GraphGA (Jensen, 2019), MARS (Xie et al., 2021), LSTM on SMILES with Hill Climbing (LSTM-HC) (Hochreiter & Schmidhuber, 1997; Brown et al., 2019), and powerful graph diffusion models: DiGress (Vignac et al., 2022), GDSS Jo et al. (2022), MOOD (Lee et al., 2023), and the state-of-the-art Graph DiT (Liu et al., 2024).

Table 1: Conditional generation of 10K molecular compounds on HIV, BBBP, and BACE.

| Tasks | Metric | Validity ↑ | Distribution Learning | | | | Condition Control | |
| | | | Coverage* ↑ | Diversity ↑ | Similarity ↑ | FCD ↓ | MAE ↓ | Accuracy * ↑ |
| | Model \ Property | | | | | | SAS | BACE, BBBP, or HIV |
| **SAS & BACE** | MOOD | 1.00 | 8/8 | 0.89 | 0.26 | 44.24 | 1.89 | 0.51 |
| | Graph GA | 1.00 | 8/8 | 0.86 | 0.98 | 7.41 | 0.96 | 0.47 |
| | Graph DiT | 0.87 | 8/8 | 0.82 | 0.88 | 7.05 | 0.40 | 0.91 |
| | STGG** | 1.00 | 8/8 | 0.82 | 0.98 | 3.82 | 0.45 | 0.95 |
| | **STGG+** ($k=1$) | 1.00 | 8/8 | 0.83 | 0.98 | 3.80 | 0.24 | 0.91 |
| | **STGG+** ($k=5$) | 1.00 | 8/8 | 0.83 | 0.98 | 3.80 | 0.18 | 0.93 |
| | **Test data** | 1.00 | 7/8* | 0.82 | 1.00 | 0.00 | 0.00† | **0.82*** |
| **SAS & BBBP** | MOOD | 0.80 | 9/10 | 0.93 | 0.17 | 34.25 | 2.03 | 0.49 |
| | Graph GA | 1.00 | 9/10 | 0.90 | 0.95 | 10.17 | 1.21 | 0.30 |
| | Graph DiT | 0.85 | 9/10 | 0.89 | 0.93 | 11.85 | 0.36 | 0.94 |
| | STGG** | 1.00 | 9/10 | 0.89 | 0.92 | 11.74 | 0.98 | 0.75 |
| | **STGG+** ($k=1$) | 1.00 | 10/10 | 0.89 | 0.94 | 9.86 | 0.47 | 0.87 |
| | **STGG+** ($k=5$) | 1.00 | 9/10 | 0.89 | 0.94 | 10.10 | 0.38 | 0.90 |
| | **Test data** | 1.00 | 10/10* | 0.88 | 1.00 | 0.00 | 0.02 | **0.81*** |
| **SAS & HIV** | MOOD | 0.29 | 29/29 | 0.93 | 0.14 | 32.35 | 2.31 | 0.51 |
| | Graph GA | 1.00 | 28/29 | 0.90 | 0.97 | 4.44 | 0.98 | 0.60 |
| | Graph DiT | 0.77 | 28/29 | 0.90 | 0.96 | 6.02 | 0.31 | 0.98 |
| | STGG** | 1.00 | 27/29 | 0.90 | 0.96 | 4.56 | 0.44 | 0.95 |
| | **STGG+** ($k=1$) | 1.00 | 27/29 | 0.90 | 0.97 | 4.08 | 0.31 | 0.88 |
| | **STGG+** ($k=5$) | 1.00 | 24/29 | 0.90 | 0.97 | 4.32 | 0.23 | 0.91 |
| | **Test data** | 1.00 | 21/29* | 0.90 | 1.00 | 0.07 | 0.02 | **0.73*** |

*The classifier from Liu et al. (2024) (used in the last column) has limited accuracy on the test set; thus, any *Property Acc.* above the **test data accuracy** is not indicative of better quality. Similarly, atom coverage is not 100% on test data; thus, any coverage above the **test set coverage** does not indicate better performance.

**STGG with categorical embedding, missing indicators, random masking, and extra symbol for compounds.

†The dataset properties are rounded to two decimals hence MAE is not exactly zero.

**Results.** The experiment results are shown in Table 1 (for the full table with more baselines, see Appendix A.8). We find that **STGG+** obtains near-perfect validity, coverage consistently higher than the test set, high diversity, and high test-set similarity. Notably, we attain the best FCD; in fact, we are the only method that matches the training data's performance, indicating that we have reached the performance cap. Regarding condition control, we achieve the best MAE on BACE and HIV, and the second-best on BBBP (very close to Graph DiT). We also obtain better performance than base STGG (with random masking and the extra symbol for compounds) on FCD and MAE, which shows that our improvements lead to lower distance in distribution and better property conditioning. We further observe that self-criticism ($k > 1$) improves property fidelity (lower MAE) while sacrificing a bit of diversity (lower coverage).

## 4.2 Out-of-distribution conditional generation

We follow the same protocol as Kwon et al. (2023). Our model is trained on Zinc250K using exact molecule weight, logP, and Quantitative Estimate of Druglikeness (QED) (Bickerton et al., 2012) as properties. For evaluation, we generate 2K candidate molecules and calculate two metrics: 1) generative efficiency, defined as the probability that the following three

conditions are satisfied at the same time: validity, uniqueness (not a duplicate), and novelty (not in train data)), and 2) the Minimum Mean Absolute Error (MinMAE) between the generated and conditioned properties (at $\pm 4$ standard-deviation). Note that for QED, the high condition value is at an impossible value of 1.2861 (the possible range is 0 to 0.948). Conditioning on impossible values is not ideal, but we choose to follow the protocol of Kwon et al. (2023) for better comparability, and it lets us verify whether our model behaves reasonably when conditioning values are difficult or impossible to achieve. We use the same train, valid, and test splits as Jo et al. (2022). Following Shao et al. (2020), we compare our model to vanilla VAE with k-annealing (BaseVAE) (Kingma & Welling, 2013; Bowman et al., 2015), ControlVAE (Shao et al., 2020), and various single-decoder (SD) and multi-decoders (MD) methods proposed by Shao et al. (2020).

Table 2: Out-of-distribution ($\mu \pm 4\sigma$) property-conditional generation of 2K molecules on Zinc250K. Generative efficiency (% of valid, novel, and unique molecules) and Minimum MAE. See Section A.10 for Mean MAE results over the top-100 molecules.

| | Generative Efficiency | | | Properties - MinMAE | | | | | |
|---|---|---|---|---|---|---|---|---|---|
| | molWt | logP | QED | molWt | | logP | | QED | |
| Condition | | | | 84 | 580 | -3.2810 | 8.1940 | 0.1778 | 1.2861* |
| *MD* | 0.49 | 0.42 | 0.47 | 9.8e−2 | 1.7e−1 | 2.0e−2 | 3.0e−4 | 1.5e−3 | 1.0e−1 |
| $MD_{dif}$ | 0.46 | 0.43 | 0.47 | 7.4e−3 | 4.7e−2 | 3.0e−4 | 5.1e−3 | 2.0e−4 | 2.6e−2 |
| $MD_{dif,col}$ | 0.46 | 0.54 | 0.44 | 1.1e−1 | 6.2e−2 | 1.3e−3 | 5.0e−4 | 6.0e−4 | 8.6e−2 |
| STGG** | 0.99 | 0.99 | 0.99 | 5.8e−2 | 7.5e−2 | 7.9e−3 | 1.9e−1 | 1.5e−2 | 8.0e−4 |
| **STGG+** $(k = 1)$ | 0.82 | 0.82 | 0.54 | 8.6e−3 | 9.1e−3 | 1.0e−4 | 1.6e−3 | 1.0e−5 | 5.1e−1 |
| **STGG+** $(k = 5)$ | 0.88 | 0.74 | 0.50 | 1.1e−3 | 1.7e−2 | 1.0e−4 | 1.6e+0 | 1.0e−4 | 5.2e−1 |
| **STGG+** $(w \sim \mathcal{U}(-0.5, 2), k = 1)$ | 0.94 | 0.92 | 0.82 | 2.1e−2 | 2.4e−2 | 1.0e−4 | 7.0e−4 | 7.0e−6 | 5.8e−3 |
| **STGG+** $(w \sim \mathcal{U}(-0.5, 2), k = 5)$ | 0.90 | 0.77 | 0.79 | 1.0e−3 | 6.1e−3 | 2.0e−7 | 2.8e−2 | 1.0e−4 | 1.2e−3 |
| **Train data** (closest sample) | - | - | - | 5.7e+1 | 7.3e+1 | 1.5e−1 | 2.0e−3 | 1.8e−2 | 8.2e−4 |

*The value is improper; we condition on 1.2861 but calculate the MAE with respect to the max QED (0.948).
**STGG with missing indicators, and random masking.

**Results.** We see in Table 2 that base STGG (with random masking) reaches the best generative efficiency (% of valid, novel, and unique molecules), but performs much worse than **STGG+** in terms of property conditioning. Our method sacrifices a small amount of generative efficiency (when compared to base STGG) in order to obtain much better property-conditioning; we see that our method generally obtains the smallest MAE. However, while the model performs optimally when using random guidance, it struggles with high guidance values when generating molecules for the impossible QED value of 1.2861. Additionally, we observe that the model performs worse with the best-of-5 when generating molecules with high logP, suggesting that the property predictor of STGG+ makes incorrect predictions for high out-of-distribution logP values. We also report the average of the top-100 molecules MAE (Top-100 MAE) instead of the top-1 MAE (MinMAE) for our STGG+ and base STGG (Table 10 in Appendix).

For Top-100 MAE, STGG+ performs much better than STGG in all cases except for high QED, where STGG is slightly better. Random guidance is helpful for high QED and logP.

## 4.3 Reward maximization

Jain et al. (2023) use reinforcement learning or GFlowNet (Bengio et al., 2023) to solve a task on the QM9 (Ramakrishnan et al., 2014) dataset. They seek to produce QM9-like molecules that maximize a reward composed of four properties: HOMO-LUMO gap (Griffith & Orgel, 1957), SAS (Ertl & Schuffenhauer, 2009), QED (Bickerton et al., 2012), and molecular weight. This reward is maximized when the HOMO-LUMO gap is as large as possible, and SAS, QED, and weight are 2.5, 1.0, and 105, respectively. We compare to Envelope QL (Yang et al., 2019), MOReinforce (Lin et al., 2022), MOA2C (Mnih et al., 2016), Multi-objective GFlowNet (MOGFN-PC) (Bengio et al., 2023). The HOMO-LUMO gap is evaluated with MXMNet (Zhang et al., 2020).

Instead of using the reward, we train a STGG+ model conditioned on the four properties directly. This shows a benefit of STGG+ our approach, since we do not need to scalarize the multi-objective reward. Since the HOMO-LUMO gap needs to be maximized there is no appropriate conditioning value. We arbitrarily set it to 0.5, which corresponds to

approximately five standard deviations (a limitation of our conditioning method, as we cannot maximize a property, only set a fixed value). The other properties are set to their optimal values: 2.5, 1.0, and 105.

Table 3: Reward maximization on QM9.

|  | Type | Data | Reward ($\uparrow$) | Diversity ($\uparrow$) |
|---|---|---|---|---|
| Envelope QL | | | 0.65 | 0.15 |
| MOReinforce | Online | 1M molecules | 0.57 | 0.47 |
| MOA2C | | | 0.61 | 0.61 |
| MOGFN-PC | | | 0.76 | 0.07 |
| STGG** | Offline | QM9 ($\sim$115K molecules) | 0.73 | 0.90 |
| **STGG+** $(k=1)$ | | | 0.78 | 0.24 |

$^{**}$STGG with missing indicators, and random masking.

**Results.** Our results are shown in Table 3. Our approach yields the highest average rewards and we obtain higher diversity than GflowNet, using $\sim$11.5% of the molecules. This makes our approach significantly more efficient. However, note that solving this task with online methods is a different setting and can be considered more difficult.

### 4.4 Hard: Small dataset of large molecules (Chromophore DB)

As a more challenging example, we explore the generation of molecules with out-of-distribution properties on Chromophore DB (Joung et al., 2020), a small dataset of around 6K molecules with an average of 35 atoms per molecule (compared to 23 atoms for Zinc250K and 9 atoms for QM9). To make the problem more realistic, we only sample 100 molecules (in the real world, chemists would decide which of those 100 molecules to synthesize based on their expert knowledge). We want to know if one of those 100 molecules has the desired out-of-distribution properties. Given the small size of the dataset, it can be useful to first pre-train on a large set of small molecules (Zinc250K) and then fine-tune on the smaller dataset of large molecules (Chromophore DB). We try this strategy (pre-train and fine-tune) in addition to training only on Chromophore DB.

Table 4: Out-of-distribution ($\mu \pm 4\sigma$) property-conditional generation of 100 molecules on Chromophore DB. Generative Efficiency (% of valid, novel, and unique molecules) and Minimum MAE. See Section A.10 for Mean MAE results.

|  | Generative Efficiency | | | | Properties - MinMAE | | | |
|---|---|---|---|---|---|---|---|---|
|  | molWt | logP | | QED | molWt | logP | | QED |
| Condition | 1538.00 | -13.63 | 28.69 | 1.24* | 1538.00 | -13.63 | 28.69 | 1.24* |
| Trained on Chromophore DB (1000 epochs) | | | | | | | | |
| **STGG+** $(k=1)$ | 0.97 | 0.33 | 0.98 | 0.59 | 9.02 | 3.30 | 0.03 | 0.30 |
| **STGG+** $(k=100)$ | 0.88 | 0.25 | 0.82 | 0.81 | 5.24 | 6.02 | 8.02 | 0.25 |
| **STGG+** $(w \sim \mathcal{U}(-0.5, 2), k=1)$ | 0.91 | 0.71 | 0.92 | 0.75 | 0.41 | 8.10 | 0.12 | 0.05 |
| **STGG+** $(w \sim \mathcal{U}(-0.5, 2), k=100)$ | 0.89 | 0.71 | 0.94 | 0.83 | 0.74 | 0.89 | 7.03 | 0.01 |
| Pre-trained on Zinc250K (50 epochs) and fine-tuned on Chromophore DB (100 epochs) | | | | | | | | |
| **STGG+** $(k=1)$ | 0.99 | 0.96 | 0.99 | 0.98 | 0.94 | 0.38 | 0.41 | 0.15 |
| **STGG+** $(k=100)$ | 1.00 | 0.96 | 0.93 | 1.00 | 2.37 | 0.35 | 0.42 | 0.09 |
| **STGG+** $(w \sim \mathcal{U}(-0.5, 2), k=1)$ | 1.00 | 0.95 | 0.97 | 1.00 | 0.47 | 0.66 | 0.01 | 0.02 |
| **STGG+** $(w \sim \mathcal{U}(-0.5, 2), k=100)$ | 1.00 | 0.92 | 0.98 | 0.99 | 13.19 | 0.45 | 0.18 | 0.01 |
| **Train data** (closest sample) | - | - | - | - | 1.40 | 9.62 | 0.17 | 0.01 |

$^{\dagger}$We removed low molWt and QED which are both impossible negative values.
$^{*}$The value of 1.24 is improper; we calculate the MAE with respect to the max QED (0.948).

**Results.** The experiment results are shown in Table 4 (see Table 11 for the top-100 MAE). Pre-training on Zinc250K generally improves performance (Efficiency and MinMAE). For most properties, random guidance with filtering ($k > 1$) leads to the closest properties. However, for high logP, we obtain better property fidelity with no filtering ($k = 1$), indicating that the model struggles with property prediction on large out-of-distribution logP values.

### 4.5   Ablations and analyses

We provide ablations for the various **STGG+** components on OOD properties for Zinc (Table 5). Since we have 6 conditions (2 per property), results can be mixed; thus, we must average over the 6 conditions. Since every condition has different scaling, to average correctly, we use as metric the relative MAE (the difference divided by absolute value of the true value; averaged over all samples and over the 6 conditions). We compare Top-100, Top-100, Top-1 Relative MAEs. We train with 3 different seeds. We show that all the components improve the performance, except randomizing the node ordering (versus using a fixed node ordering) which has no significant impact. Variance across different seeds is low.

We provide visualizations of generated molecules in Appendix A.9.1, A.9.2, A.9.3. The visualizations show that STGG+ can generate molecules that satisfy OOD conditions while maintaining validity.

Table 5: Ablation for the out-of-distribution ($\mu \pm 4\sigma$) property-conditional generation on Zinc. Average Relative MAE over the 6 conditions (low/high molWt, logP, QED). The numbers correspond to Mean (Standard-deviation) over 3 seeds.

| | Top-100 MAE | Top-10 MAE | Top-1 MAE (MinMAE) |
|---|---|---|---|
| **STGG+** ($w \sim \mathcal{U}(-0.5, 2), k = 1$) | 0.0247 (0.0029) | 0.0040 (0.0005) | 0.0010 (0.0001) |
| without randomize-order | 0.0252 (0.0021) | 0.0040 (0.0009) | 0.0009 (0.0004) |
| without MLP (1-layer) | 0.0260 (0.0029) | 0.0046 (0.0007) | 0.0012 (0.0005) |
| without the property-prediction loss | 0.0264 (0.0017) | 0.0052 (0.0004) | 0.0012 (0.0001) |
| without improved architecture | 0.0348 (0.0009) | 0.0074 (0.0003) | 0.0014 (0.0003) |
| different guidance ($w \sim \mathcal{U}(0.5, 2)$) | 0.0974 (0.0081) | 0.0567 (0.0169) | 0.0332 (0.0186) |
| without guidance | 0.1010 (0.0082) | 0.0834 (0.0192) | 0.0719 (0.0272) |
| with fixed guidance ($w = 1.5$) | 0.1085 (0.0022) | 0.0930 (0.0123) | 0.0867 (0.0192) |
| without standardized properties | 0.1309 (0.0287) | 0.0322 (0.0062) | 0.0190 (0.0045) |
| with self-criticism ($k = 2$) | 0.0207 (0.0022) | 0.0029 (0.0002) | 0.0004 (0.0004) |
| with self-criticism ($k = 5$) | 0.0203 (0.0039) | 0.0023 (0.0007) | 0.0006 (0.0004) |
| with self-criticism ($k = 10$) | 0.0269 (0.0113) | 0.0035 (0.0018) | 0.0007 (0.0004) |

We note that self-criticism marginally improves top-100 MAE (t(4) = 1.90, p = .06) and significantly improve top-1 MAE (t(4) = 2.52, p = .03) and top-10 MAE (t(4) = 3.54, p = .01).

### 4.6   Limitations

Molecule validity as measured by RDKit is widespread in machine learning, which is why we use it, but it is not a full picture of chemical validity. A molecule could be unstable or implausible by physical laws not implemented in RDKit; the field of generating *synthesizable* molecules still has many open questions. Another limitation is that many properties (e.g., molecule weight, logP) are not especially interesting from a chemistry perspective. We use them because they are widespread in the literature due to their ease of use through existing tools. This is why we included datasets with more complex properties, such as HOMO-LUMO gap and the categorical features for HIV, BBBP, and BACE. These either require slow simulations or fast machine learning models that may be inaccurate, especially in OOD settings. We believe that establishing new benchmarks with more complex properties is essential to the progress of the field. The property predictor of our approach may not be as accurate as property predictors made explicitly for this task, meaning our method may not always select the best molecules out of $k$ choices, particularly in OOD scenarios (see A.6 for the analysis); we found this to be the case for large OOD logP conditioning values.

## 5   Conclusion

In this paper, we demonstrated that with specific techniques, optimization, and architectural improvements, spanning tree-based graph generation (STGG) can be leveraged to generate high-quality and diverse molecules conditioned on both *in-distribution* and *out-of-distribution* properties. Our method achieves equal or superior performance on validity, novelty, uniqueness, closeness in distribution, and conditioning fidelity compared to competing approaches. Using

fewer molecules than required by online methods (RL/GFlowNet), we also obtain high multi-property-reward molecules in a one-shot manner from a pre-trained model.

**Broader Impact**

The ability to generate novel molecules with desired properties has the potential to significantly advance research in drug discovery, materials science, and sustainability. Tools such as ours could accelerate the design of therapeutics, catalysts, and electronic materials, contributing to improved health outcomes and more efficient technologies. However, as with many generative models, there is a dual-use risk: the same methods could, in principle, be misused to design toxic, addictive, or otherwise harmful compounds. While our work is intended solely for beneficial scientific applications, we acknowledge that unrestricted access to powerful molecular design tools requires careful consideration of security and ethical safeguards. Future work should explore mechanisms for responsible deployment, including dataset curation, access control, and collaboration with domain experts and regulatory bodies to ensure that such technologies are used safely and ethically.

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

# A   Appendix

## A.1   1D vs 2D representations

There are many ways to represent molecules in the context of molecular generation. The most popular methods are autoregressive models on 1D strings and diffusion (Song & Ermon, 2019; Ho et al., 2020; Song et al., 2020) models on 2D graphs. We highlight the main distinction between the two representations below in the context.

Let $D$ be the size of the training dataset, $n$ be the number of atoms in a given molecule, $d$ is the embedding size, and $b$ is the number of bond types.

Diffusion models on 2D graphs:

- $G = (X, A)$ where the vertices $X$ contains the list of atoms (size: $[n, d]$) and $A$ is the adjacency matrix of the edges (size: $[n, n, b]$) for each bond type.

- $A$ is an extremely sparse matrix with many zero elements

- Input space is $\mathcal{O}(nd + bn^2)$; unless using low-rank projections, the number of parameters must scale proportionally to this amount

- Typically use diffusion models (or related methods) given the large number of steps it would take to generate $X$ and $A$ autoregressively

- Equivariant Graph Neural Networks (E-GNNs) are generally used to ensure a unique representation for a given molecule

- Although it has a single representation per molecule, multiple random noises per graph are needed due to diffusion; thus, sample complexity is $\mathcal{O}(Dn_{noise})$

Autoregressive models on 1D strings:

- $X$ (size: $[L, d]$) is a string containing the molecule where $L$ is proportional to $n$

- The string starts from a random atom and traverses the 2D molecular graph

- Input space is $\mathcal{O}(nd)$; this makes it efficient to process

- Typically use autoregressive models (e.g., Transformers) as it scales well

- We can either 1) fix the ordering in some way to make representation unique, or 2) use random orderings as data augmentation with a non-unique representation for a given molecule; thus, sample complexity is $\mathcal{O}(Dn_{augments})$

As can be seen, both methods have similar sample complexity, but 1D strings are much more compressed representations, leading to less space and parameters and, thus, increased potential for scalability. Furthermore, recent results show that 1D strings are as competitive as 2D molecular graph methods for unconditional molecule generation Jang et al. (2023); Fang et al. (2023) and property prediction Yüksel et al. (2023). In the end, both representations contain as much information. Thus, the choice is a matter of preference. 1D strings are easier to scale and can make good use of the power of Transformers; hence, we focus on this type of representation.

## A.2   Spanning Tree compared to other 1D string-based representations

The most popular choice of string-based representation is SMILES (Weininger, 1988). SMILES is extremely versatile, allowing the representation of any molecule. However, for the purpose of generative models, trying to generate SMILES strings directly can quickly lead to many invalid molecules. Graph-based diffusion methods encounter the same issue. Recently, methods have been created to prevent the creation of invalid molecules: Spanning Tree (Ahn et al., 2021) and SELFIES (Krenn et al., 2022). Below, we describe in detail the differences between all three methods.

SMILES:

- Massive vocabulary allows the representation of every aspect of molecules

- There are many ways of representing a single molecule

- Begin-branch token "(" to deviate from the main path and close branch token ")"

- Pointer token $i$ to indicate both the beginning and end of rings

SELFIES:

- Restricted SMILES vocabulary

- Prevent invalid molecules through a carefully designed context-free grammar:

  – Atoms and bonds are combined into single tokens (with other aspects such as charge and number of hydrogen atoms) so that we cannot have an atom without a bond and a bond without an atom

  – Hard-designed rules for maximum valencies of specific elements (slightly more permissible than octet rule, but cannot handle every case)

  – Keep track of valencies; ignore future tokens in the current branch if there is not enough valency left and reduce bond order if needed

  – There is an open-branch token Branch-$i$ and close-ring token ring-$i$ where $i$ specifies the number of future tokens in the branch and how many backward steps (in tokens) are needed to reach the ring closure; this ensures that all branches and tokens are not left opened

Spanning-Tree:

- Restricted SMILES vocabulary

- Begin and end branch tokens, with ring start and $i$-th ring end tokens

- Similarity-based output layer to determine the probabilities of ring ends and input embedding injection for how many rings are opened

- Prevent invalid molecules through masking of tokens before softmax:

  – Masking of invalid tokens due to impossible valencies (atoms, ring-start, and branch-start when not enough valency is left) based on the valencies of the training data

  – Masking of invalid next tokens (atom after atom, bond after bond, ring-$i$ end when there are less than $i$ ring start tokens)

  – Force branch ending through masking when getting too close to maximum sequence length to prevent unfinished molecules (new)

As can be seen, SMILES has such a large vocabulary that each molecule can be represented in completely different ways, and its main problem for generative models is that many token choices lead to invalid molecules (e.g., two bonds, incorrect valencies, unfinished branches, or rings, etc.).

SELFIES prevents invalid molecules through its smart, context-free grammar. However, work by Gao et al. (2022) and Ghugare et al. (2023) found that while SELFIES prevent invalid molecules, it makes exploration more difficult and reduces the performance of generative models (in terms of obtaining high-reward samples, i.e., molecules with desired properties). A significant challenge for the generative models based on SELFIES is the need to pre-define the number of tokens contained in a branch (a deviation from the main path in a 1D string) and count backward the number of tokens required to reach the beginning of the ring (starting from the end). This requires extensive planning and counting by the generative model.

On the other hand, Spanning-tree use clever masking of incorrect tokens to prevent invalid molecules and doing so does not require the model to do significant planning-in-advance and counting when selecting the next token (including the $i$-th ring end tokens which require no counting due to the similarity-based prediction). Note that Wang et al. (2024) and Pandey et al. (2024) also use a similar masking to the one devised by Ahn et al. (2021) to improve validity, thus it can be applied to other representations.

### A.3 Datasets details and canonicalization

QM9 (Ramakrishnan et al., 2014) has 21 atom tokens: CH3, C, O, CH2, CH, NH, N, N-, NH+, OH, NH2, F, NH3+, O-, NH2+, N+, C-, CH-, NH3, OH2, CH4. The maximum length is 37. The dataset has 133886 molecules with around 10% of the molecules in the test set and 5% in the validation set.

Zinc250K (Sterling & Irwin, 2015) has 34 atom tokens: CH3, C, CH, N, S, CH2, O, NH, NH+, NH2, NH2+, NH3+, OH, Cl, O-, N-, F, Br, N+, S-, I, SH, P, NH-, O+, OH+, S+, CH2-, CH-, SH+, PH, PH+, P+, PH2. The maximum length is 136. The dataset has 250k molecules with around 10% of the molecules in the test set and 5% in the validation set.

BBBP (Wu et al., 2018) has 31 atom tokens: CH, C, F, CH2, N, S, CH3, O, OH, NH2, Cl, NH, OH2, Br, O-, N+, Na, Cl-, H+, C-, Na+, NH+, NH3+, Br-, P, N-, SH, CH2-, CH-, I, B. The maximum length is 186. The dataset has 862 molecules with around 20% of the molecules in the test set and 20% in the validation set.

BACE (Wu et al., 2018) has 20 atom tokens: F, C, N, CH, NH+, CH2, NH2, O, Cl, S, CH3, NH, OH, NH2+, Br, O-, NH3+, N+, N-, I. The maximum length is 161. The dataset has 1332 molecules with around 20% of the molecules in the test set and 20% in the validation set.

HIV (Wu et al., 2018) has 76 atom tokens: CH3, C, O, CH2, N, NH2, CH, N+, NH2+, I, NH, Br, Se, OH, S, O-, Br-, SH, Cl, I-, S+, Zn-2, OH+, N-, NaH, PH, Ir-3, Cl-, NH3, F, P, BrH, C-, Co-2, Cu-4, As, B-2, Sn, ClH, Rh-4, O+, S-, Pt, Fe-2, B, U+2, Pd-2, Fe-3, Pt-2, Pt+2, Si, P+, IH2, Fe, SiH, Cl+3, Ge, NH+, Zr, K+, AlH3-, IH, KH, Mn+, Fe-4, Cu-3, Ni-4, LiH, Co-3, Pd-3, Fe+2, Ga-3, CH2-, U, Mn, Co-4. The maximum length is 193. The dataset has 2372 molecules with around 20% of the molecules in the test set and 20% in the validation set.

Chromophore DB (Joung et al., 2020) has 46 atom tokens: CH, C, N, CH3, CH2, O, N+, B-, F, S, OH, NH, Cl, NH2, P, O+, Si, O-, Se, C+, B, Br, I, NH+, NH2+, N-, S+, SiH, C-, Na, Sn, NH3+, S-, Si-, P-, Cl+3, I-, BH3-, P+, BH, CH4, NH-, SH, Ge, Te, Na+. The maximum length is 511. The dataset has 6810 molecules with around 5% of the molecules in the test set and 5% in the validation set.

Note that we base the maximum length on the largest SMILES string after being transformed with the Spanning tree tokenizer.

### A.3.1 Canonicalization

Similar to STGG (Ahn et al., 2021), we use explicit Hydrogen atoms (with no implicit Hydrogen atom) in the tokens. This is an arbitrary choice. After generation, we always transform back to canonical SMILES using RDKit (Landrum et al., 2024). Note that RDKit may change the number of Hydrogen atoms based on its own rule-set. All our molecule figures are based on RDKit so they reflect the molecules after SMILES canonicalization by RDKit.

Here is an example below.

Training SMILES: C[C@@]12C=CC(=O)C=C1CC[C@@H]
1[C@@H]3CC[C@](O)(C(=O)COP(=O)([O-])[O-])[C@]3(C)C[C@@H](O)[C@H]12

STGG tokenized: [bos]C-C[bor][bor]-C=C-C(=O)(-C=C(-[eor0])(-C-C-CH[bor]-CH[bor]-C-C-C(-O)(-C(=O)(-C-O-P(=O)(-O-)(-O-)))(-C(-[eor3])(-C)(-C-CH(-O)(-CH(-[eor1])(-[eor2]))))))[eos]

Canonical SMILES: CC12C=CC(=O)C=C1CCC1C2C(O)
CC2(C)C1CCC2(O)C(=O)COP(=O)([O-])[O-]

### A.3.2 Any-property masking

For any-property masking, we show an example below. Assume that there are $T = 5$ properties. Each time we sample a training molecule, we choose a random number $t$ of properties to mask uniformly between 0 and $T = 5$. Assuming that $t = 3$, we create the masking vector: [1, 1, 1, 0, 0, 0]. Then, we randomly shuffle the masking vector, leading to: [1, 0, 0, 1, 0, 1]. Then, we mask the properties with a masking value of 1.

### A.4 Hyperparameters

The original STGG (Ahn et al., 2021) used the AdamW optimizer (Loshchilov & Hutter, 2017; Kingma & Ba, 2014) with $\beta_1 = 0.9$, $\beta_2 = 0.999$, no weight decay, a fixed learning rate of 2e-4 and batch-size 128. The Transformer architecture had 3 layers, dropout 0.1, 8 attention heads, and embedding size 1024. They processed only one property with a single linear layer.

**STGG+** uses the AdamW optimizer with $\beta_1 = 0.9$, $\beta_2 = 0.95$, and weight decay 0.1. The Transformer architecture has 3 layers, no dropout, 16 attention heads, SwiGLU (Hendrycks & Gimpel, 2016; Shazeer, 2020) with expansion scale of 2, no bias term (Chowdhery et al., 2023), Flash Attention (Dao et al., 2022; Dao, 2023), RMSNorm (Zhang & Sennrich, 2019), Rotary embeddings (Su et al., 2024), residual-path weight initialization (Radford et al., 2019). When not using random guidance, we use classifier-free guidance with a guidance parameter $w = 1.5$, where $w = 1$ means no guidance.

For QM9 (Ramakrishnan et al., 2014), we train for 50 epochs with batch size 512, learning rate 1e-3, max length 150. For Zinc250K (Sterling & Irwin, 2015), we train for 50 epochs with batch size 512, learning rate 1e-3, max length 250. For HIV, BACE, and BBBP (Wu et al., 2018), we train for 10K epochs (same as done by Liu et al. (2024)), since these are small datasets, with batch size 128, learning rate 2.5e-4, max length 300.

For Chromophore DB (Joung et al., 2020), we train for 1000 epochs with batch size 128, learning rate 2.5e-4, max length 600. For the pre-training on Zinc250K and fine-tuning on Chromophore-DB: we pre-train with batch size 512, learning rate 1e-3, and max length 600 for 50 epochs and fine-tune with batch size 128, learning rate 2.5e-4, and max length 600 for 100 epochs.

We generally use 1 to 4 A-100 GPUs to train the models. Training takes less than a day. Note that we use a higher max length than the data max length (generally around 25-50%) to ensure that we can adequately generate molecules with out-of-distribution properties that could be bigger than usual. Generating 10K molecules takes a few minutes with 1 GPU.

For pretraining and then fine-tune, there are two ways to preprocess the properties: we can either standardize them with respect to the pre-training or the fine-tuning datasets. Standardizing with respect to the pre-training dataset can lead to extreme values in the fine-tuning (e.g., 4 standard deviation in Chromophore's MolWt is 15 standard-deviation in Zinc250K's MolWt). Hereby, to reduce the gap between pre-trained and fine-tuned conditioning values, we preprocess the properties by standardizing with respect to the fine-tune dataset properties during both pre-training and fine-tuning.

### A.5 Alternative architectures considered

In this work, we enhance the Transformer architecture used by Ahn et al. (2021) using recent developments in Large Language Models (LLMs). Although powerful, the Transformer architecture with self-attention (Vaswani et al., 2017) is quadratic in context length, which means that the time and memory increase significantly when dealing with long-context length.

In addition to improvements on Transformer, new architectures such as Mamba (Gu & Dao, 2023), Hyena, (Poli et al., 2023) or RWKV (Peng et al., 2023) have appeared, which are sub-quadratic with respect to context-length, allowing them to handle long-context length better. We initially considered some of these architectures to improve inference speed. However, it is hard to synthesize and manufacture molecules of substantial sizes. Thus, the context length is generally quite limited (e.g., the largest molecule on Chromophore has 511 tokens, while modern LLMs have a context length of at least 4096). As long as the context length is less or equal to 2048, FlashAttention (Dao et al., 2022) is fast enough that there is no inference speed benefit for using Mamba (Gu & Dao, 2023).

## A.6 Property prediction

Table 6: Property prediction on the test set using **STGG+** or a Random Forest (Breiman, 2001) predictor/classifier using the Morgan Fingerprint (Morgan, 1965) as done by Gao et al. (2022).

| | | Accuracy | Mean squared error (MSE) | | | | | |
|---|---|---|---|---|---|---|---|---|
| Task | Method | HIV | QED | MolWt | logP | SAS | SCS | Gap |
| QM9 | STGG+ | - | 0.0010 | 0.0018 | 0.0012 | - | - | - |
| QM9 | Random Forest | - | 0.2665 | 0.6124 | 0.2014 | - | - | - |
| Zinc250K | STGG+ | - | 0.0008 | 0.0005 | 0.0005 | - | - | - |
| Zinc250K | Random Forest | - | 0.4077 | 0.4209 | 0.3907 | - | - | - |
| HIV | STGG+ | 0.8463 | - | - | - | 0.0268 | 0.0216 | - |
| HIV | Random Forest | 0.7263 | - | - | - | 0.3605 | 0.4672 | - |
| BACE | STGG+ | 0.9551 | - | - | - | 0.0126 | 0.0070 | - |
| BACE | Random Forest | 0.8165 | - | - | - | 0.1773 | 0.3948 | - |
| BBBP | STGG+ | 0.8743 | - | - | - | 0.0354 | 0.0314 | - |
| BBBP | Random Forest | 0.8057 | - | - | - | 0.3152 | 0.4740 | - |
| QM9 (Reward) | STGG+ | - | - | 0.0009 | 0.0005 | 0.0021 | - | 0.0032 |
| QM9 (Reward) | Random Forest | - | - | 0.6122 | 0.2015 | 0.2114 | - | 0.1459 |

We analyze the property prediction performance of the self-critic on out-of-distribution (OOD) molecules (Figure 3). To do so, we generated 10,000 molecules using our trained STGG+ conditioned on property values randomly drawn from the Zinc distribution. We condition on all three properties at the same time (QED, MolWt, logP). We then compute the ground truth property values for each generated molecule using RDKit as in our other experiments and predict the property values with the self-critic. We measure the absolute error between the ground truth and predicted property values and plot how the error changes depending on how far the ground truth is from the distribution mean. The results show that the error gradually increases with the distribution shift indicating the estimated limits of self-critic's reliability. Note that in Table 6 the errors are computed differently: for in-distribution (based on the test set molecules in Zinc instead of the generated ones), so the errors are much lower in that case.

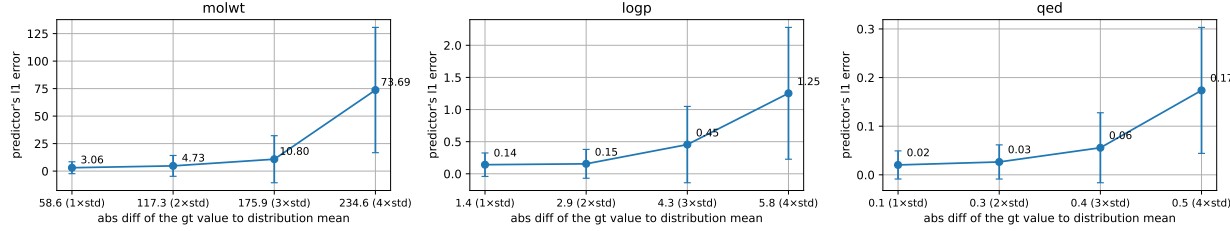

Figure 3: Analysis of the property predictor performance in the OOD regime. The values on the $y$ axis are the mean errors for the corresponding bin on the $x$ axis (e.g. at point $2 \times$ std we compute the mean error for all molecules with ground truth property values between $1 \times$ std and $2 \times$ std). The error bar shows standard deviation (std) within the bin.

## A.7 Unconditional Generation

Table 7: Molecular graph generation performance on QM9.

| Method | Valid (%) (↑) | Unique (%) (↑) | Novel (%) (↑) | FCD (↓) | Scaf. (↑) | SNN (↑) | Frag. (↑) |
|---|---|---|---|---|---|---|---|
| Domain-agnostic graph generative models | | | | | | | |
| EDP-GNN | 47.52 | **99.25** | 86.58 | 2.680 | 0.3270 | 0.5265 | 0.8313 |
| GraphAF | 74.43 | 88.64 | 86.59 | 5.625 | 0.3046 | 0.4040 | 0.8319 |
| GraphDF | 93.88 | 98.58 | **98.54** | 10.928 | 0.0978 | 0.2948 | 0.4370 |
| GDSS | 95.72 | 98.46 | 86.27 | 2.900 | 0.6983 | 0.3951 | 0.9224 |
| DiGress | 98.19 | 96.67 | 25.58 | 0.095 | 0.9353 | 0.5263 | 0.0023 |
| DruM | 99.69 | 96.90 | 24.15 | 0.108 | **0.9449** | 0.5272 | 0.9867 |
| GraphARM | 90.20 | - | - | 1.220 | - | - | - |
| GEEL | **100.0** | 96.08 | 22.30 | 0.089 | 0.9386 | 0.5161 | 0.9891 |
| Molecule-specific generative models | | | | | | | |
| CharRNN | 99.57 | - | - | **0.087** | 0.9313 | 0.5162 | 0.9887 |
| CG-VAE | **100.0** | - | - | 1.852 | 0.6628 | 0.3940 | 0.9484 |
| MoFlow | 91.36 | 98.65 | 94.72 | 4.467 | 0.1447 | 0.3152 | 0.6991 |
| **STGG** | **100.0** | 96.76 | 72.73 | 0.585 | 0.9416 | **0.9998** | **0.9984** |
| Unconditional (masking all properties) | | | | | | | |
| **STGG+** | **100.0** | 97.17 | 74.41 | 0.089 | 0.9265 | 0.5179 | 0.9877 |
| Conditional (using test properties) | | | | | | | |
| **STGG+** (k=1) | **100.0** | 97.63 | 75.99 | 0.134 | 0.8906 | 0.5004 | 0.9799 |
| **STGG+** (k=5) | **100.0** | 96.86 | 74.18 | 0.166 | 0.9050 | 0.5039 | 0.9860 |

Table 8: Molecular graph generation performance on Zinc250K.

| Method | Valid (%) (↑) | Unique (%) (↑) | Novel (%) (↑) | FCD (↓) | Scaf. (↑) | SNN (↑) | Frag. (↑) |
|---|---|---|---|---|---|---|---|
| Domain-agnostic graph generative models | | | | | | | |
| EDP-GNN | 63.11 | 99.79 | **100.00** | 16.737 | 0.0000 | 0.0815 | 0.0000 |
| GraphAF | 68.47 | 98.64 | 99.99 | 16.023 | 0.0672 | 0.2422 | 0.5348 |
| GraphDF | 90.61 | 99.63 | **100.00** | 33.546 | 0.0000 | 0.1722 | 0.2049 |
| GDSS | 97.01 | 99.64 | **100.00** | 14.656 | 0.0467 | 0.2789 | 0.8138 |
| DiGress | 94.99 | 99.97 | 99.99 | 3.482 | 0.4163 | 0.3457 | 0.9679 |
| DruM | 98.65 | 99.97 | 99.98 | 2.257 | 0.5299 | 0.3650 | 0.9777 |
| GraphARM | 88.23 | - | - | 16.260 | - | - | - |
| GEEL | 99.31 | 99.97 | 99.89 | 0.401 | 0.5565 | 0.4473 | 0.9920 |
| Molecule-specific generative models | | | | | | | |
| CharRNN | 96.95 | - | - | 0.474 | 0.4024 | 0.3965 | **0.9988** |
| CG-VAE | **100.0** | - | - | 11.335 | 0.2411 | 0.2656 | 0.8118 |
| MoFlow | 63.11 | 99.99 | **100.00** | 20.931 | 0.0133 | 0.2352 | 0.7508 |
| **STGG** | **100.0** | 99.99 | 99.89 | **0.278** | **0.7192** | **0.4664** | 0.9932 |
| Unconditional (masking all properties) | | | | | | | |
| **STGG+** | **100.0** | 99.99 | 99.94 | 0.395 | 0.5657 | 0.4316 | 0.9925 |
| Conditional (using test properties) | | | | | | | |
| **STGG+** (k=1) | **100.0** | **100.0** | 99.98 | 0.514 | 0.5302 | 0.4099 | 0.9917 |
| **STGG+** (k=5) | **100.0** | **100.0** | **100.0** | 0.562 | 0.5491 | 0.4176 | 0.9909 |

## A.8 Full Table of conditional generation on HIV, BBBP, and BACE

Table 9: Full table: Conditional generation of 10K molecular compounds on HIV, BBBP, and BACE.

| Tasks | Model | Validity ↑ | Distribution Learning | | | | Condition Control | |
|---|---|---|---|---|---|---|---|---|
| | | | Coverage* ↑ | Diversity ↑ | Similarity ↑ | Distance ↓ | Synthe. MAE ↓ | Property Acc.* ↑ |
| Synth. & BACE | DiGress | 0.351 | 8/8 | 0.886 | 0.694 | 24.656 | 2.068 | 0.506 |
| | DiGress v2 | 0.355 | 8/8 | 0.881 | 0.703 | 25.327 | 2.337 | 0.511 |
| | GDSS | 0.288 | 4/8 | 0.876 | 0.271 | 46.754 | 1.642 | 0.504 |
| | MOOD | 0.995 | 8/8 | 0.890 | 0.259 | 44.239 | 1.885 | 0.506 |
| | Graph DiT | 0.867 | 8/8 | 0.824 | 0.875 | 7.046 | 0.400 | 0.913 |
| | Graph GA | 1.000 | 8/8 | 0.859 | 0.981 | 7.410 | 0.963 | 0.469 |
| | MARS | 1.000 | 8/8 | 0.834 | 0.883 | 6.792 | 1.012 | 0.518 |
| | LSTM-HC | 0.997 | 8/8 | 0.815 | 0.798 | 17.559 | 0.921 | 0.582 |
| | JTVAE-BO | 1.000 | 6/8 | 0.668 | 0.728 | 30.470 | 0.992 | 0.463 |
| | STGG** | 1.000 | 8/8 | 0.824 | 0.979 | 3.824 | 0.453 | 0.949 |
| | **STGG+**$(k=1)$ | 1.000 | 8/8 | 0.829 | 0.979 | 3.796 | 0.238 | 0.912 |
| | **STGG+**$(k=5)$ | 1.000 | 8/8 | 0.826 | 0.979 | 3.802 | 0.178 | 0.926 |
| | **Train data** | 1.000 | 8/8 | 0.819 | 0.981 | 3.837 | 0.003$^\dagger$ | 0.991 |
| | **Test data** | 1.000 | **7/8**$^*$ | 0.824 | 1.000 | 0.000 | 0.002$^\dagger$ | **0.817**$^*$ |
| Synth. & BBBP | DiGress | 0.696 | 9/10 | 0.910 | 0.681 | 18.692 | 2.366 | 0.654 |
| | DiGress v2 | 0.689 | 9/10 | 0.911 | 0.634 | 19.450 | 2.269 | 0.653 |
| | GDSS | 0.622 | 3/10 | 0.842 | 0.267 | 39.944 | 1.379 | 0.504 |
| | MOOD | 0.801 | 9/10 | 0.927 | 0.172 | 34.251 | 2.028 | 0.490 |
| | Graph DiT | 0.847 | 9/10 | 0.886 | 0.933 | 11.851 | 0.355 | 0.942 |
| | Graph GA | 1.000 | 9/10 | 0.895 | 0.951 | 10.166 | 1.208 | 0.302 |
| | MARS | 1.000 | 8/10 | 0.864 | 0.770 | 10.979 | 1.225 | 0.519 |
| | LSTM-HC | 0.999 | 8/10 | 0.888 | 0.893 | 16.390 | 0.997 | 0.559 |
| | JTVAE-BO | 1.000 | 5/10 | 0.746 | 0.582 | 33.575 | 1.162 | 0.496 |
| | STGG** | 1.000 | 9/10 | 0.891 | 0.916 | 11.736 | 0.982 | 0.754 |
| | **STGG+**$(k=1)$ | 1.000 | 10/10 | 0.888 | 0.937 | 9.859 | 0.466 | 0.867 |
| | **STGG+**$(k=5)$ | 1.000 | 9/10 | 0.887 | 0.936 | 10.101 | 0.381 | 0.900 |
| | **Train data** | 1.000 | 8/10 | 0.883 | 0.957 | 9.890 | 0.017$^\dagger$ | 0.996 |
| | **Test data** | 1.000 | **10/10**$^*$ | 0.880 | 0.998 | 0.000 | 0.018$^\dagger$ | **0.806**$^*$ |
| Synth. & HIV | DiGress | 0.438 | 22/29 | 0.919 | 0.856 | 13.041 | 1.922 | 0.534 |
| | DiGress v2 | 0.505 | 24/29 | 0.919 | 0.848 | 13.400 | 1.593 | 0.533 |
| | GDSS | 0.693 | 4/29 | 0.782 | 0.103 | 45.342 | 1.252 | 0.483 |
| | MOOD | 0.288 | 29/29 | 0.928 | 0.136 | 32.352 | 2.314 | 0.511 |
| | Graph DiT | 0.766 | 28/29 | 0.897 | 0.958 | 6.022 | 0.309 | 0.978 |
| | Graph GA | 1.000 | 28/29 | 0.899 | 0.966 | 4.442 | 0.984 | 0.604 |
| | MARS | 1.000 | 26/29 | 0.876 | 0.652 | 7.289 | 0.969 | 0.646 |
| | LSTM-HC | 0.999 | 13/29 | 0.909 | 0.915 | 7.466 | 0.948 | 0.674 |
| | JTVAE-BO | 1.000 | 3/29 | 0.806 | 0.417 | 41.977 | 1.236 | 0.485 |
| | STGG** | 1.000 | 27/10 | 0.899 | 0.961 | 4.558 | 0.442 | 0.950 |
| | **STGG+**$(k=1)$ | 1.000 | 27/29 | 0.896 | 0.970 | 4.075 | 0.314 | 0.876 |
| | **STGG+**$(k=5)$ | 1.000 | 24/29 | 0.897 | 0.9700 | 4.317 | 0.229 | 0.905 |
| | **Train data** | 1.000 | 27/29 | 0.895 | 0.970 | 4.019 | 0.018$^\dagger$ | 0.999 |
| | **Test data** | 1.000 | **21/29**$^*$ | 0.895 | 0.998 | 0.074 | 0.015$^\dagger$ | **0.726**$^*$ |

$^*$The classifier from Liu et al. (2024) (used in the last column) has limited accuracy on the test set; thus, any *Property Acc.* above the **test data accuracy** is not indicative of better quality. Similarly, atom coverage is not 100% on test data; thus, any coverage above the **test set coverage** does not indicate better performance.

$^{**}$STGG with categorical embedding, missing indicators, random masking, and extra symbol for compounds.

$^\dagger$The dataset properties are rounded to two decimals hence MAE is not exactly zero.

## A.9 Min-MAE or Max-Reward molecules generated by STGG+

### A.9.1 Zinc OOD

Figure 4: Conditioning on molWt=580.00

Figure 7: Conditioning on logP=-3.2810

Figure 5: Conditioning on molWt=84.0008

Figure 8: Conditioning on QED=1.2861

Figure 6: Conditioning on logP=8.1940

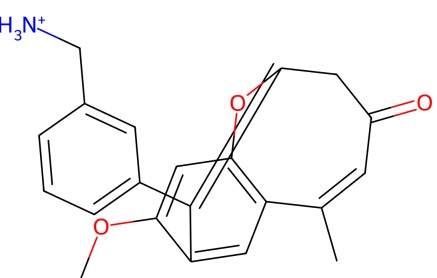

Figure 9: Conditioning on QED=0.1778 (which means low drug-likeness and less "chemical beauty" (Bickerton et al., 2012))

### A.9.2 QM9 Reward Maximization

Figure 10: Best QM9 reward maximization molecules

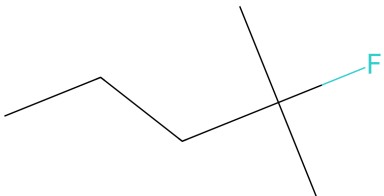

### A.9.3 Chromophore OOD

Figure 11: Conditioning on molWt=1538.00

Figure 12: Conditioning on logP=28.6915

Figure 13: Conditioning on logP=-13.6292

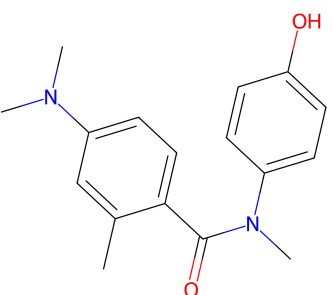

Figure 14: Conditioning on QED=1.2355

### A.10 OOD Tables with Mean MAE over the top-100 molecules

Table 10: Out-of-distribution ($\mu \pm 4\sigma$) property-conditional generation of 2K molecules on Zinc250K.

| | Properties - top-100 Mean MAE | | | | | |
|---|---|---|---|---|---|---|
| | molWt | | logP | | QED | |
| Condition | 84 | 580 | -3.2810 | 8.1940 | 0.1778 | 1.2861* |
| STGG** | 18.248 | 5.559 | 1.204 | 1.548 | 0.206 | 0.022 |
| **STGG+**$(k = 1)$ | 0.790 | 1.389 | 0.018 | 0.900 | 0.003 | 0.561 |
| **STGG+**$(k = 5)$ | 1.289 | 1.503 | 0.021 | 3.710 | 0.003 | 0.571 |
| **STGG+**$(w \sim \mathcal{U}(-0.5, 2), k = 1)$ | 1.533 | 2.088 | 0.040 | 0.285 | 0.005 | 0.060 |
| **STGG+**$(w \sim \mathcal{U}(-0.5, 2), k = 5)$ | 1.285 | 1.104 | 0.022 | 0.803 | 0.004 | 0.042 |

*The value is improper; we condition on 1.2861 but calculate the MAE with respect to the maximum QED (0.948).

**STGG with missing indicators, and random masking.

Table 11: Out-of-distribution ($\mu \pm 4\sigma$) property-conditional generation of 100 molecules on Chromophore DB. We removed the low molWt and QED which are both impossible negative values.

| | Properties - Mean MAE | | | |
|---|---|---|---|---|
| | molWt | logP | | QED |
| Condition | 1538.00 | -13.63 | 28.69 | 1.24* |
| Trained on Chromophore DB (1000 epochs) | | | | |
| **STGG+**$(k = 1)$ | 256.6 | 11.1 | 5.1 | 0.6 |
| **STGG+**$(k = 100)$ | 562.3 | 11.0 | 16.3 | 0.5 |
| **STGG+**$(w \sim \mathcal{U}(-0.5, 2), k = 1)$ | 805.6 | 15.4 | 11.1 | 0.5 |
| **STGG+**$(w \sim \mathcal{U}(-0.5, 2), k = 100)$ | 609.5 | 8.8 | 14.3 | 0.2 |
| Pre-trained on Zinc250K (50 epochs) and fine-tuned on Chromophore DB (100 epochs) | | | | |
| **STGG+**$(k = 1)$ | 294.9 | 8.4 | 6.1 | 0.5 |
| **STGG+**$(k = 100)$ | 401.9 | 5.6 | 13.1 | 0.4 |
| **STGG+**$(w \sim \mathcal{U}(-0.5, 2), k = 1)$ | 543.0 | 14.6 | 12.7 | 0.5 |
| **STGG+**$(w \sim \mathcal{U}(-0.5, 2), k = 100)$ | 416.5 | 6.1 | 13.0 | 0.2 |

*The value of 1.24 is improper; we calculate the MAE with respect to the maximum QED (0.948).

In Table 11, STGG+ with pre-training and fine-tuning generally performs slightly better than regular training. Random guidance is helpful for high QED.

## A.11 Algorithms

---

**Algorithm 1** STGG+ Training

---

**Require:** Dataset $\mathcal{D} = \{(x_i, y_i)\}$ where $x_i$ is a molecule and $y_i \in \mathbb{R}^D$ are its properties
**Require:** Transformer model $f_\theta$

1: **while** not converged **do**
2:     Sample batch $(x_1, \ldots, x_B)$ and properties $(y_1, \ldots, y_B)$
3:     **for** each molecule $x_i$ in batch **do**
4:         Tokenize $x_i$ into sequence $(t_1, \ldots, t_L)$
5:         Mask a random subset of $m$ properties from $y_i = (y_{i1}, ..., y_{iD})$, where $m \sim \text{Uniform}(0, D)$
6:         Compute $(h_1, \ldots, h_L)$, where $h_j \leftarrow f_\theta(t_j | y_i, t_1, ..., t_{j-1})$
7:         Compute the cross-entropy loss $\mathcal{L}_{\text{CE}}$
8:         Compute the auxiliary property prediction loss $\mathcal{L}_{\text{prop}} = \frac{1}{2}\|f_\theta^{\text{pred}}(h_L) - y_i\|_2^2$
9:     Update $\theta$ using gradient descent on $\mathcal{L} = \mathcal{L}_{\text{CE}} + \lambda\mathcal{L}_{\text{prop}}$

---

**Algorithm 2** STGG+ Sampling with Self-Criticism

---

**Require:** Target properties $y_{\text{target}} \in \mathbb{R}^T$
**Require:** Guidance strength $w$, number of candidates $K$, max length $L_{\text{max}}$
**Require:** Transformer model $f_\theta$

1: Generate $K$ candidate molecules:
2: **for** $k = 1$ to $K$ **do**
3:     Initialize the sequence; $t_1 \leftarrow$ [BOS]
4:     Sample guidance scale $w \sim \mathcal{U}(0.5, 2)$ (optional; otherwise w=1 means no guidance)
5:     **for** $j = 2$ to $L_{\text{max}}$ **do**
6:         Compute conditional logits $z_c = f_\theta(t_j | y_{\text{target}}, t_1, ..., t_{j-1})$
7:         Compute unconditional logits $z_u = f_\theta(t_j | \emptyset, t_1, ..., t_{j-1})$
8:         Apply Classifier-Free Guidance (CFG): $z = wz_c + (1 - w)z_u$
9:         Mask invalid tokens (valency violations, syntax errors, ring overflow, etc.)
10:         Sample the next token $t_j \sim \text{Softmax}(z)$
11:         **if** $t_j \leftarrow$ [EOS] **then**
12:             **break**
13:     Store the molecule $s_k$
14: Self-criticism:
15: **for** each candidate $s_k$ **do**
16:     Process $s_k$ through $f_\theta$ with empty properties to predict $\hat{y}_k$
17:     Compute distance $d_k = \|\hat{y}_k - y_{\text{target}}\|_2^2$
18: **return** $s_j$ where $j \leftarrow \arg\min_k d_k$ {Select the best candidate}

---

### A.12 Masking

Here we describe the original STGG masking algorithm and the improved **STGG+** masking algorithm.

---

**Algorithm 3** STGG Masking algorithm

---

**Require:** maximum number of rings $ring_{max}$

1: **if** token is an atom **then**
2:     $allowed_{next} = [bonds, branch_{start}, ring_{start}]$
3:     **if** all branches are closed **then**
4:         $allowed_{next}.append([EOS])$
5:     **else**
6:         $allowed_{next}.append([branch_{end}])$
7: **if** token is a bond **then**
8:     $allowed_{next} = [atoms]$
9:     **for** $i \leftarrow 1$ to $ring_{max}$ **do**
10:         **if** $ring^i$ has not been closed **then**
11:             $allowed_{next}.append([ring_{end}^i])$
12: **if** token is a branch start **then**
13:     $allowed_{next} = [bonds]$
14: **if** token is a branch end **then**
15:     $allowed_{next} = [branch_{start}]$
16:     **if** all branches are closed **then**
17:         $allowed_{next}.append([EOS])$
18:     **else**
19:         $allowed_{next}.append([branch_{end}])$
20: **if** token is a ring start **then**
21:     $allowed_{next} = [bonds, branch_{start}, ring_{start}]$
22:     **if** all branches are closed **then**
23:         $allowed_{next}.append([EOS])$
24:     **else**
25:         $allowed_{next}.append([branch_{end}])$
26: **if** token is a ring end **then**
27:     $allowed_{next} = []$
28:     **if** all branches are closed **then**
29:         $allowed_{next}.append([EOS])$
30:     **else**
31:         $allowed_{next}.append([branch_{end}])$
32: **if** Beginning Of Sentence (BOS) **then**
33:     $allowed_{next} = [atoms]$
34: **if** End Of Sentence (EOS) **then**
35:     $allowed_{next} = []$
36: Apply the valency mask (remove illegal tokens from $allowed_{next}$ based on the valency of the atoms)
37: Mask every token not included in $allowed_{next}$ before sampling the next token

---

---

**Algorithm 4 STGG+** Masking algorithm

---

**Require:** maximum number of rings $ring_{max}$
**Require:** maximum length of a sequence $MAX_{LEN}$
 1: **if** token is an atom **then**
 2:    $allowed_{next} = [bonds, branch_{start}]$
 3:    **if** this is the second double-bond in a row within a branch **then**
 4:       $allowed_{next}.remove([bond_{double}])$
 5:    **if** we have made less than $ring_{max}$ rings **then**
 6:       $allowed_{next}.append([ring_{start}])$
 7:    **if** all branches are closed **then**
 8:       $allowed_{next}.append([EOS])$
 9:       **if** empty bonds (.) are allowed (for compounds such as $[Na+].[Cl-]$) **then**
10:          $allowed_{next}.append([bond_{empty}])$
11:    **else**
12:       $allowed_{next}.append([branch_{end}])$
13: **if** token is a bond **then**
14:    $allowed_{next} = [atoms]$
15:    **for** $i \leftarrow 1$ to $ring_{max}$ **do**
16:       **if** $ring^i$ has not been closed **then**
17:          $allowed_{next}.append([ring^i_{end}])$
18: **if** token is a branch start **then**
19:    $allowed_{next} = [bonds]$
20:    **if** this is the second double-bond in a row within a branch **then**
21:       $allowed_{next}.remove([bond_{double}])$
22: **if** token is a branch end **then**
23:    $allowed_{next} = [branch_{start}]$
24:    **if** all branches are closed **then**
25:       $allowed_{next}.append([EOS])$
26:       **if** empty bonds (.) are allowed **then**
27:          $allowed_{next}.append([bond_{empty}])$
28:    **else**
29:       $allowed_{next}.append([branch_{end}])$
30: **if** token is a ring start **then**
31:    $allowed_{next} = [bonds, branch_{start}]$
32:    **if** this is the second double-bond in a row within a branch **then**
33:       $allowed_{next}.remove([bond_{double}])$
34:    **if** we have made less than $ring_{max}$ rings **then**
35:       $allowed_{next}.append([ring_{start}])$
36:    **if** all branches are closed **then**
37:       $allowed_{next}.append([EOS])$
38:       **if** empty bonds (.) are allowed **then**
39:          $allowed_{next}.append([bond_{empty}])$
40:    **else**
41:       $allowed_{next}.append([branch_{end}])$
42: **if** token is a ring end **then**
43:    $allowed_{next} = []$
44:    **if** all branches are closed **then**
45:       $allowed_{next}.append([EOS])$
46:       **if** empty bonds (.) are allowed **then**
47:          $allowed_{next}.append([bond_{empty}])$
48:    **else**
49:       $allowed_{next}.append([branch_{end}])$
50: **if** Beginning Of Sentence (BOS) or Empty bond (.) **then**
51:    $allowed_{next} = [atoms]$
52: **if** End Of Sentence (EOS) **then**
53:    $allowed_{next} = []$
54: When getting too close to $MAX_{LEN}$, if possible, remove atom, bonds, ring-start, branch-start tokens from $allowed_{next}$
55: Apply the valency mask (remove illegal tokens from $allowed_{next}$ based on the valency of the atoms)
56: Mask every token not included in $allowed_{next}$ before sampling the next token

---

