# OpenReview forum: "Any-Property-Conditional Molecule Generation with Self-Criticism using Spanning Trees"
_TMLR — Accepted by TMLR_

### Review · Reviewer_jLEz · 2025-10-06

**Summary Of Contributions:**

The paper's primary contribution is advancing the

1) Spanning Tree-based Graph Generation (STGG) framework from a purely unconditional model into a state-of-the-art, multi-property conditional molecule generator called STGG+. The authors achieve this through several clever innovations. They introduce a training strategy where properties are

2) randomly masked , which allows the final model to generate molecules based on any combination of target properties without being retrained. Furthermore, they incorporate a novel


3) "self-criticism" mechanism by training the model to also predict a molecule's properties ; during generation, this allows the model to create several candidates and then intelligently select the best one. By combining these ideas with an upgraded, modern Transformer architecture and an advanced sampling technique called

4) "random guidance," they create a model that excels at generating high-quality, valid molecules that meet specific criteria, even for challenging out-of-distribution targets.

**Audience:**

Yes

**Audience Explanation:**

This research is interesting because it offers a powerful, efficient, and highly flexible tool for designing custom molecules from scratch, which has significant real-world implications across several scientific and technological fields.

The most direct application is in accelerating the search for new medicines. Researchers can use this model to generate novel molecules with a specific therapeutic profile.

**Broader Impact Concerns:**

The submission includes a "Broader Impact" section that acknowledges the core ethical implication of the work: its dual-use potential. The statement correctly identifies that the technology could be used for beneficial purposes, such as developing new drugs and materials, but also notes that "harmful molecules could also be discovered"

My main concern is that this statement, while accurate, is insufficiently addressed and overly brief. The potential for misuse of a powerful molecular design tool warrants a more thorough discussion. The work enables the generation of novel molecules with specific, targeted properties, which could theoretically be used by malicious actors to design new toxins, chemical weapons, or illicit substances.

**Claims And Evidence:**

Yes

**Claims Explanation:**

Yes, the claims made in the submission are well-supported by accurate, convincing, and clear evidence presented throughout the paper.

The central claim that STGG+ achieves state-of-the-art performance is convincingly supported across three distinct and challenging tasks.

1) In-Distribution Generation: In Table 1, the authors compare STGG+ against powerful recent models, including the graph diffusion model Graph DiT. The results clearly show STGG+ achieving the best performance on the Fréchet ChemNet Distance (FCD) metric and the lowest (best) Mean Absolute Error (MAE) on the BACE and HIV datasets, demonstrating superior property conditioning and distributional similarity.

2) Out-of-Distribution (OOD) Generation: For the more difficult task of generating molecules with extreme properties, Table 2 shows that STGG+ obtains a much smaller property error (MinMAE) than competing methods and the base STGG model. This provides strong evidence of its effectiveness in novel chemical space exploration.

3) Reward Maximization: The paper demonstrates in Table 3 that STGG+ achieves the highest average reward on a multi-objective task, surpassing online methods like GFlowNet. The evidence is particularly convincing as it also highlights the model's superior data efficiency, using only about 11.5% of the molecules required by the online approaches.

The claim that the novel components of STGG+ are crucial to its success is clearly validated through a detailed ablation study. Table 5 systematically removes or alters key components of the model, such as the improved architecture, the property-prediction loss (which enables self-criticism), and the random guidance strategy. In each case, the model's performance degrades, providing direct and convincing evidence that each of these contributions is beneficial and collectively responsible for the final model's high performance.

**Requested Changes:**

The following are recommendations that would improve the paper's completeness and practical relevance

1) Provide a Proxy Metric for Synthesizability. The authors correctly state that the validity metric used, as measured by RDKit, is not a full picture of chemical validity and that generating synthesizable molecules remains an open question. To better address this, I suggest reporting a standard proxy for synthesizability, such as the Synthetic Accessibility (SA) score, for the molecules generated in the out-of-distribution and reward maximization tasks. This would provide stronger evidence that the model generates plausible molecules, not just computationally valid but overly complex ones.

2) Include a Deeper Analysis of the Self-Critic's Failure Modes. The paper transparently notes that its internal property predictor may be less accurate than specialized predictors and can struggle in out-of-distribution (OOD) scenarios, which can cause the model to select a suboptimal molecule. To better quantify this limitation, it would be beneficial to add a brief analysis showing how the predictor's accuracy degrades as the conditioning values become more extreme. This would offer readers valuable insight into the model's reliability limits.

3) Explicitly Demonstrate "Any-Property" Conditioning. A key contribution is the model's ability to condition on any subset of properties without retraining, enabled by random masking of properties during training. While the experiments on multi-property datasets implicitly support this claim, the work would be more convincing if this flexibility were demonstrated explicitly. For example, the authors could add a small experiment in the appendix using a model trained on the HIV, BACE, or BBBP datasets  to show its performance when conditioned on only one or two properties, rather than all of them. This would provide direct validation for this powerful and practical feature.

---

> ### Author Response · Authors · 2025-11-14
> **response**
>
> We thank the reviewer for recognizing our contribution and for their suggestions. We  address their concerns below. The paper has been updated accordingly, with all changes highlighted in red in the revised manuscript.
>
> > Provide a Proxy Metric for Synthesizability. The authors correctly state that the validity metric used, as measured by RDKit, is not a full picture of chemical validity and that generating synthesizable molecules remains an open question. To better address this, I suggest reporting a standard proxy for synthesizability, such as the Synthetic Accessibility (SA) score, for the molecules generated in the out-of-distribution and reward maximization tasks. This would provide stronger evidence that the model generates plausible molecules, not just computationally valid but overly complex ones.
>
> We agree that synthesizability is extremely important. Currently the experiments on the three datasets in Table 1 condition on SA score. We did not include the SA score in the other experiments since it is not part of the pre-existing benchmarks. In principle, we can include SA as one of the properties that we condition on and thus ask for a specific SA score along with the other properties. We believe that this would stray a bit too far from the original benchmark though, so we choose to leave this to future work.
>
>
> > 2. Include a Deeper Analysis of the Self-Critic's Failure Modes. The paper transparently notes that its internal property predictor may be less accurate than specialized predictors and can struggle in out-of-distribution (OOD) scenarios, which can cause the model to select a suboptimal molecule. To better quantify this limitation, it would be beneficial to add a brief analysis showing how the predictor's accuracy degrades as the conditioning values become more extreme.
>
> Thank you for the suggestion. We have added a Figure in appendix (A.6) showing the changes in property-predictor accuracy with respect to the conditioning value. This shows how and when performance starts degrading significantly (starting at 3 standard-deviations away from the mean).
>
> > 3. Explicitly Demonstrate "Any-Property" Conditioning. A key contribution is the model's ability to condition on any subset of properties without retraining, enabled by random masking of properties during training. While the experiments on multi-property datasets implicitly support this claim, the work would be more convincing if this flexibility were demonstrated explicitly. For example, the authors could add a small experiment in the appendix using a model trained on the HIV, BACE, or BBBP datasets to show its performance when conditioned on only one or two properties, rather than all of them. This would provide direct validation for this powerful and practical feature.
>
> We thank the reviewer for this suggestion. Our OOD experiments in Tables 2 and 4 already provide an explicit demonstration of the any-property conditioning: although the model is trained jointly on three properties, at test time we mask two properties and condition on a single property at approximately ±4 standard deviations from the training mean. We have clarified this more explicitly in the manuscript.
>
> > 4. The submission includes a "Broader Impact" section that acknowledges the core ethical implication of the work: its dual-use potential. The statement correctly identifies that the technology could be used for beneficial purposes, such as developing new drugs and materials, but also notes that "harmful molecules could also be discovered" My main concern is that this statement, while accurate, is insufficiently addressed and overly brief. The potential for misuse of a powerful molecular design tool warrants a more thorough discussion. The work enables the generation of novel molecules with specific, targeted properties, which could theoretically be used by malicious actors to design new toxins, chemical weapons, or illicit substances.
>
>
> Thank you for the suggestion, we have revamped our broader impact section:
>
> “The ability to generate novel molecules with desired properties has the potential to significantly advance research in drug discovery, materials science, and sustainability. Tools such as ours could accelerate the design of therapeutics, catalysts, and electronic materials, contributing to improved health outcomes and more efficient technologies. However, as with many generative models, there is a dual-use risk: the same methods could, in principle, be misused to design toxic, addictive, or otherwise harmful compounds. While our work is intended solely for beneficial scientific applications, we acknowledge that unrestricted access to powerful molecular design tools requires careful consideration of security and ethical safeguards. Future work should explore mechanisms for responsible deployment, including dataset curation, access control, and collaboration with domain experts and regulatory bodies to ensure that such technologies are used safely and ethically.”

---

### Review · Reviewer_XUrS · 2025-10-15

**Summary Of Contributions:**

This paper introduces STGG+, an advanced framework for any-property-conditional molecule generation, building upon the Spanning Tree-based Graph Generation (STGG) model. It combines modern Transformer techniques (RMSNorm, rotary embeddings, FlashAttention-2, and SwiGLU) with new strategies such as random property masking, self-criticism via auxiliary property prediction, and randomized classifier-free guidance (CFG). These improvements enable flexible conditioning on arbitrary molecular properties and enhance generalization to out-of-distribution (OOD) settings.

The model is evaluated on multiple benchmarks, including QM9, Zinc250K, BBBP, BACE, HIV, and Chromophore DB. The results show near-perfect molecular validity, strong property control, and improved diversity compared to prior baselines such as Graph DiT and MOOD. The authors also perform extensive ablation studies, confirming that each component contributes meaningfully to the overall performance.

# Strengths

- The integration of self-criticism and random property masking introduces genuine novelty. Allowing the model to evaluate its own outputs represents a creative step forward in molecular generation.

- The adoption of state-of-the-art Transformer design principles makes the framework scalable, efficient, and aligned with contemporary LLM architecture trends.

- The model is tested across multiple datasets and task types, including OOD property conditioning and reward maximization. The empirical scope is broad and thorough.

# Weaknesses

- The evaluation relies on RDKit-based validity metrics, which do not guarantee chemical realism or synthetic feasibility. There is no expert or simulation-based validation of generated molecules.

- The use of basic properties such as QED, logP, and SAS limits the scope of the evaluation. These metrics are convenient but not always chemically meaningful.

- The self-evaluation predictor is trained jointly with the generator, which may lead to bias and inaccuracy, especially in OOD property regions.

- Some baselines differ in training paradigms (offline vs. online), making comparisons less direct. Computational costs and training efficiency are not fully discussed.

**Audience:**

Yes

**Audience Explanation:**

The topic is about the generative models, bioinformatics, molecules, MCTS, which are widely discussed topics in machine learning community.

**Claims And Evidence:**

Yes

**Claims Explanation:**

n/a

**Requested Changes:**

See weaknesses

---

> ### Author Response · Authors · 2025-11-14
> **response**
>
> We thank the reviewer for recognizing our contributions. We address their concerns below. The paper has been updated accordingly, with all changes highlighted in red in the revised manuscript.
>
> > The evaluation relies on RDKit-based validity metrics, which do not guarantee chemical realism or synthetic feasibility. There is no expert or simulation-based validation of generated molecules.
>
> > The use of basic properties such as QED, logP, and SAS limits the scope of the evaluation. These metrics are convenient but not always chemically meaningful.
>
> > The self-evaluation predictor is trained jointly with the generator, which may lead to bias and inaccuracy, especially in OOD property regions.
>
> We fully  agree with these points; please note that we already discuss them in the limitations section of the manuscript.
>
> > Some baselines differ in training paradigms (offline vs. online), making comparisons less direct. Computational costs and training efficiency are not fully discussed.
>
> Details on training computation costs are found in A.4. Generation speed is extremely fast (a few minutes even for as much as 10K molecules); we now mention this in A.4.

---

### Review · Reviewer_gJMm · 2025-11-05

**Summary Of Contributions:**

The authors propose an extension of an existing approach for generating molecules, and aim to satisfy molecule properties specified by the user. As a limitation of the existing model, the authors identify that it often generates invalid molecules. They mitigate this shortcoming by introducing several improvements, including how a partial molecule can be extended (to prevent moves that are likely to end up generating an invalid molecule), and providing the model with a self-criticism module. The authors also draw on a range of recent advances in deep learning and integrate several approaches for improving training. In an empirical evaluation, the authors demonstrate that their proposed model outperforms several baselines, achieving state-of-the-art performance in several commonly used benchmarks. However, in some cases, their restrictions for guiding molecule generation limit the diversity of generated molecules compared to the original approach they extend.

I found the writing of the paper clear, and the authors' ideas easy to follow. However, I also believe that the work could be somewhat more self-contained. Specifically, the authors make references to many recent deep learning advances, which they integrate into their model, but they do not explain how those recent concepts work, what their expected benefit is, or why it makes sense to use them. I believe that readers who are very familiar with molecule generation or closely follow rather technical developments in deep learning will probably have no trouble following along. However, for a general machine learning audience, it may be more difficult to grasp why each of the components was chosen.

### Strengths
- This work considers an important problem, namely generating molecules with certain properties.
- The proposed method integrates recent deep learning advances and outperforms the chosen baselines on commonly used benchmark datasets.

### Weaknesses
- The work could be more self-contained. Specifically, to address a broader audience, the authors could explain in more detail how the model they extend (STGG) works and why they make each specific architectural choice (section 3.1).
- The authors make claims that are not supported with evidence: (1) in the conclusion, the authors claim that their approach is "extremely efficient and fast", however, they do not explain further what that means, for example by comparing empirical runtimes with baselines; (2) the authors claim that their model's self-criticism "significantly improves the quality of its generated molecules", however, they do not use any statistical significance test to support this claim.

**Audience:**

Yes

**Audience Explanation:**

Yes, I believe this is the case. At least those who work on molecule generation or related topics (transformers and technologies powered by transformers, such as large language models) would be interested in the findings. The present work is probably interesting to at least some readers because it combines several recent developments in deep learning to push the state-of-the-art in molecule generation.

**Broader Impact Concerns:**

The authors have included a "Broader Impact" statement, where they acknowledge the possibility that their work could be used to generate "harmful molecules". I do not have any concerns regarding their statement, and do not believe that it needs to be changed.

**Claims And Evidence:**

No

**Claims Explanation:**

The authors claim and show that their method outperforms current baselines for molecule generation. However, as mentioned above, there are claims that are not sufficiently supported by evidence. Specifically:

1. In the conclusion, the authors claim that their approach is "extremely efficient and fast", however, they do not explain further what that means, for example, by comparing empirical runtimes with baselines
2. The authors claim that their model's self-criticism "significantly improves the quality of its generated molecules", however, they do not use any statistical significance test to support this claim.

**Requested Changes:**

### Critical points
- The statement regarding that STGG+ is "extremely efficient and fast" needs to be supported by evidence. This could be done, for example, by showing empirical runtimes (if this is what the authors mean, otherwise, they need to clarify what they mean by "fast").
- If the authors wish to claim significance, they need to support the statement that the "best-out-of-k strategy significantly improves the quality of its generated molecules" with statistical significance tests.
- In section 3.3, the authors mention that they extend the vocabulary of STGG, which enables "the model to solve a broader range of problems". However, I do not believe that the authors explain what problems STGG+ can solve that STGG cannot solve. I believe the authors should clarify this, so the reader may understand in what ways STGG+ is more powerful than STGG.
- The concept of "extreme out-of-distribution properties" comes up several times, but it is not explained what exactly this means. I believe this should be explained (to be honest, though, I am not an expert in molecule generation, so if this is a common term and the reader can be expected to know what this means, you can discard this comment).
- I believe it would be useful to say some words about the real-world readiness of STGG+. Specifically, in this paper, the authors considered "6 conditions (2 per property)". Is that a realistic setting? How many settings can STGG+ handle? Is there some upper limit? How does STGG+ scale to situations with many properties? How much (more) training data is required per additional property?


### Not critical, but would strengthen the paper
- I believe it would strengthen the paper if the authors could explain in at least some detail why they make the architectural choices mentioned in section 3.1. As it stands now, I believe this is merely a list of many recently proposed works, but it does not help the reader understand *why* the authors made these choices.
- I find the example for missing features in section 3.2 confusing and did not find that it helped my understanding. Perhaps the authors can rephrase.
- For categorical values, the authors mention that they add an extra category to represent missing values. But in the end, they need to be encoded with floats anyway, right? Or are they one-hot encoded vectors? Perhaps a brief explanation would be useful.
- The authors mention that they "mask a random subset of $t$ properties, where $t$ is chosen uniformly between $0$ and the total number of properties". Is it realistic that a subset of $t$ random features could be missing? Or does it not matter what a realistic setting is because the model must be able to deal with any missing subset of features?
- The authors mention that their data-centric approach "allows [them] to represent complex structures", which sounds a bit as if a non-data-centric approach would not be capable of doing that. Is that what is meant here?
- It sounds a little strange to say that "we generally maintain 100% validity". If it is always 100%, then it's not necessary to say "generally". And if there are situations with less than 100%, then one cannot say that it is "generally 100%". This should probably be rephrased.
- What do you mean when you say that the classifier-free guidance "[directs] the model more towards the conditional model's direction while pushing it away from the unconditional model's direction"? Does "the model" refer to what STGG+ has currently learned? And "the conditional model" is the true, but unknown model? In any case, I feel like there are too many "directions" in the sentence, which was what confused me.
- It seems a bit strange to point out, in the main text, that a certain paper had been selected for an oral presentation at NeurIPS 2024. Perhaps this should be removed.
- In section 4.3, GflowNet is mentioned as one of the baselines, but does not appear in Table 3.
- Why are there no baselines used in section 4.4, Table 4?


### Minor points
- I believe it should be "we cannot assume ~the~ plausible values for the missing features"
- Since it is rare that the "problematic situation of incompletely-generated samples" occurs, I am wondering whether the additional masking step to prevent this adds much? Could one not instead simply increase $k$ and just assume that, since problematic situations are rare, everything will be fine?
- You mention that you "mask the creation of rings", did you mean "the token of rings"?
- Redundant "molecules molecules properties" in section 3.6.
- In section 4, the abbreviation "OoD" is used while the rest of the paper uses "OOD".

---

> ### Author Response · Authors · 2025-11-14
> **response (1/2)**
>
> We thank the reviewer for their thorough review and for recognizing our contributions. We address their concerns below. The paper has been updated accordingly, with all changes highlighted in red in the revised manuscript.
>
> > to address a broader audience, the authors could explain in more detail how the model they extend (STGG) works and why they make each specific architectural choice (section 3.1).
>
> Thank you for the suggestion. We have added detailed explanations of the Transformer architecture and clarified the motivation for each of the 7 improvements.  We also reorganized the section on 1D representations to better contextualize STGG and explain why it is needed (sections 2.2.1 SMILES, 2.2.2 SELFIES, 2.2.3 STGG ).
>
> > In the conclusion, the authors claim that their approach is "extremely efficient and fast", however, they do not explain further what that means
>
> We have removed the claim that it was "extremely efficient and fast". In fact the main benefit of STGG+ is improved generation metrics rather than faster and more efficient generation. The original efficiency remark was intended to emphasize that STGG+ does not require retraining when the target combination of properties changes, whereas RL methods trained with a scalarized reward (a linear combination of properties) typically need a new model for each new combination. To avoid overstatement, we no longer highlight STGG+ as “extremely efficient and fast.”
>
> > The authors claim that their model's self-criticism "significantly improves the quality of its generated molecules", however, they do not use any statistical significance test to support this claim.
>
> We have removed the word “significantly” from our original claim and added one-sided t-test p-values in Section 4.5 (Ablations and Analyses). The revised text now reads:  “We note that self-criticism marginally improves top-100 MAE (t(4) = 1.90, p = .06) and significantly improve top-1 MAE (t(4) = 2.52, p = .03) and top-10 MAE (t(4) = 3.54, p = .01).”
>
> > In section 3.3, the authors mention that they extend the vocabulary of STGG, which enables "the model to solve a broader range of problems". However, I do not believe that the authors explain what problems STGG+ can solve that STGG cannot solve. I believe the authors should clarify this, so the reader may understand in what ways STGG+ is more powerful than STGG.
>
> In Section 3.3, we now clarify concretely what additional problems STGG+ can address compared to STGG. The extended vocabulary introduces a token for ionic bonds, which allows us to represent ionic compounds such as salt (NaCl) and vitamin B12 that cannot be encoded with the original STGG. In addition, STGG+ automatically infers valency from the data, enabling it to handle unconventional structures such as hypervalent molecules (with more than eight valence electrons). This broadened representational capacity allows STGG+ to work with a wider range of molecular structures, including ionic compounds and atypical molecules, and we have reworded the corresponding sentence to explicitly mention these cases.
>
> > The concept of "extreme out-of-distribution properties" comes up several times, but it is not explained
>
> By “extreme out-of-distribution properties,” we refer to conditioning on target property values that lie roughly four standard deviations or more away from the mean of the training distribution. We have revised the text to define this explicitly and to clarify that it corresponds to rare or unseen property values that are far outside the range observed during training.
>
> > it would be useful to say some words about the real-world readiness of STGG+. Specifically, in this paper, the authors considered "6 conditions (2 per property)". Is that a realistic setting? How many settings can STGG+ handle? Is there some upper limit? How does STGG+ scale to situations with many properties? How much (more) training data is required per additional property?
>
> In practice, the number of conditions that STGG+ can handle depends strongly on the specific task and the properties of interest, so providing a general upper bound is difficult. At this stage, we believe that improving the quality and chemical relevance of the labels (a direction we are actively working on) is more critical than simply increasing the number of properties. A detailed study of how STGG+ scales with the number of properties—including the trade-off between added properties and required training data—is an interesting and valuable direction, but we view it as beyond the scope of the present paper, which focuses on establishing the basic viability of the conditional generation setting for molecules.
>
> > I find the example for missing features in section 3.2 confusing and did not find that it helped my understanding. Perhaps the authors can rephrase.
>
> We agree and rephrased section 3.2, including the example, which hopefully improves the clarity of that section.

---

> ### Author Response · Authors · 2025-11-14
> **response (2/2)**
>
> > For categorical values, the authors mention that they add an extra category to represent missing values. But in the end, they need to be encoded with floats anyway, right? Or are they one-hot encoded vectors?
>
> As mentioned in section 3.2: “Each categorical feature is then processed individually using a linear embedding”. A one-hot encoding followed by a linear transform is mathematically equivalent to a linear embedding. We make this a bit clearer in the text now. We also reworded “linear embedding” to “embedding layer”.
>
> > authors mention that they "mask a random subset of  properties, where  is chosen uniformly between  and the total number of properties". Is it realistic that a subset of  random features could be missing? Or does it not matter what a realistic setting is because the model must be able to deal with any missing subset of features?
>
> It matters a lot for experimentation even if the training data has no missing values. For example, imagine you have properties A,B,C,D and the chemist tells you that they are very important so you condition on all 4 for training. But then, they may ask you to generate molecules with A=1, B=2, disregarding C,D. And then later, they change their mind and want to add C=2 as an extra condition. The idea is that the masking allows you to generate at different subsets of condition without retraining.
>
> > authors mention that their data-centric approach "allows them to represent complex structures", which sounds a bit as if a non-data-centric approach would not be capable of doing that. Is that what is meant here?
>
> Molecules have very specific rules around the atoms that they can contain. Metals, and compounds such as salt (NaCl) are not molecules. By using the data to automatically extract the valency rules, we can handle structures that are not molecules (NaCl,  compounds, etc.) directly from the data without needing to manually check and determine what are the rules.
>
> > "we generally maintain 100% validity". This should probably be rephrased.
>
> It's because there is extremely rarely 1 molecule that is invalid out of 10,000. This could arise from weird exotic cases of molecules that cannot be read by the RDKIT software. We rephrased it to “we maintain near perfect validity”.
>
> > What do you mean when you say that the CFG directs the model more towards the conditional model's direction while pushing it away from the unconditional model's direction"? Does "the model" refer to what STGG+ has currently learned? And "the conditional model" is the true, but unknown model?
>
> This is best understood from the formula. See Figure 2 the softmax equation at the top. When w is greater than 1, we push toward the conditional model f(next-token|properties) and push away from the unconditional model f(next-token|masked(properties)).  We now mention this in the text to clarify because we agree that it is hard to understand without the equation formula context.
>
> > It seems a bit strange to point out that a certain paper had been selected for an oral presentation at NeurIPS 2024. Perhaps this should be removed.
>
> We agree, we removed it.
>
> > In section 4.3, GflowNet is mentioned as one of the baselines, but does not appear in Table 3.
>
> MOGFN-PC is Multi-objective GFlowNet (GFN stands for GFlowNet).
>
> > Why are there no baselines used in section 4.4, Table 4?
>
> There are no baselines in Section 4.4 because this challenging setting was not considered in the original VAE work, and we were unable to reproduce their results: we contacted the original authors, but the code for their experiments is no longer available. As a result, a direct comparison in this specific regime is not feasible. Instead, Table 4 is intended primarily to (i) compare STGG+ against STGG and (ii) demonstrate that STGG+ performs well on more challenging settings with larger molecules. We note that we do include extensive comparisons to baselines in several other experimental settings where running or reproducing the baselines is tractable.
>
> > Since it is rare that the "problematic situation of incompletely-generated samples" occurs, I am wondering whether the additional masking step to prevent this adds much? Could one not instead simply increase  and just assume that, since problematic situations are rare, everything will be fine?
>
> It is rare, but if the conditioning is OOD (for example a bizarre never seen combination: tiny elephant when normally elephants are huge), the model could get stuck trying to make an impossible molecule leading to something that keeps growing and never ends. While rare, we want to be sure that it does not happen because it could for example lead to zero valid molecules and make generation much slower.
>
> > should be "we cannot assume the plausible values for the missing features"
>
> > did you mean "the token of rings"?
>
> > Redundant "molecules molecules properties"
>
> > abbreviation "OoD" is used while the rest of the paper uses "OOD".
>
> Good observations, we fixed those errors.

---

### Decision · Action_Editor_E6Ui · 2025-12-04

**Recommendation:** Accept as is

**Audience:**

Yes

**Audience Explanation:**

Paper is relevant for small molecule generation tasks.

**Claims And Evidence:**

Yes

**Claims Explanation:**

The paper does good science and tests the claims it makes.